# Differentiable Structure Learning and Causal Discovery for General Binary Data

**Chang Deng**[†*]     **Bryon Aragam**[†]
[†]Booth School of Business, University of Chicago, Chicago, IL 60637

## Abstract

Existing methods for differentiable structure learning in discrete data typically assume that the data are generated from specific structural equation models. However, these assumptions may not align with the true data-generating process, which limits the general applicability of such methods. Furthermore, current approaches often ignore the complex dependence structure inherent in discrete data and consider only linear effects. We propose a differentiable structure learning framework that is capable of capturing *arbitrary* dependencies among discrete variables. We show that although general discrete models are unidentifiable from purely observational data, it is possible to characterize the complete set of compatible parameters and structures. Additionally, we establish identifiability up to Markov equivalence under mild assumptions. We formulate the learning problem as a single differentiable optimization task in the most general form, thereby avoiding the unrealistic simplifications adopted by previous methods. Empirical results demonstrate that our approach effectively captures complex relationships in discrete data.

## 1   Introduction

Causal relationships are often represented by directed acyclic graphs (DAGs), where nodes correspond to variables and directed edges indicate cause-effect links [33, 42, 35]. Learning a DAG from observational data (structure learning, also known as causal discovery) is a well-known NP-complete problem [8, 10]. Recent advances have recast this combinatorial search as a differentiable constrained program, enabling the use of gradient-based methods for structure learning [54, 55]. In this approach, the adjacency matrix of the graph is treated as a continuous variable and one optimizes a score (e.g. negative log-likelihood) subject to a differentiable acyclicity constraint that ensures the solution is a valid DAG [54, 5]. Early work in this area has largely been limited to continuous data and relied on specific structural equation models (SEMs)—for example, linear or nonlinear functions with additive Gaussian noise [36]. While such assumptions facilitate tractable optimization, they may misrepresent the true generative process for non-Gaussian or discrete data, leading to biased or inconsistent structure learning.

Many real-world datasets involve binary or other discrete variables (e.g. presence/absence of a condition, binary genetic markers, survey responses), whose complex dependency structures are not well handled by methods designed for continuous data. Traditional Gaussian [53, 17] or continuous assumptions often break down in these cases. However, only a few works have attempted to extend differentiable structure learning to discrete data. These existing approaches typically impose specific parametric forms on the data-generating process. For instance, a recent method [49] for discrete differentiable structure learning assumes a generalized linear SEM. Moreover, the theoretical identifiability guarantees of these methods rely on particular distributional constraints that may not hold for general discrete data. These gaps call for a more flexible approach that can capture the rich dependence patterns in discrete data without relying on untested generative assumptions.

---

[*]{changdeng,bryon}@chicagobooth.edu

39th Conference on Neural Information Processing Systems (NeurIPS 2025).

Outside the differentiable paradigm, decades of work have produced both *score-based* [e.g. 9] and *constraint-based* [e.g. 41] algorithms for causal discovery. Many discrete methods also impose additional simplifications (additive-noise models [34], hidden compact representations [7], linear effects [27], latent Gaussian variables [1]), which may not apply to discrete data. Even the most recent differentiable approaches evaluated on discrete data lack a formal justification: For instance, Bello et al. [5] apply continuous optimization methods to discrete data under a logistic regression model, although the underlying identifiability and theory for such a model has yet to be worked out. These observations underscore the need for a general, theoretically-sound framework for differentiable structure learning in discrete data.

In this work, we propose a general differentiable structure learning framework for discrete data, which captures arbitrary dependencies without assuming a specific data-generating process. We first show that, in this general setting, DAGs are non-identifiable from observational data alone. Nevertheless, we characterize the complete set of compatible structures and parameters. To select among these, we adopt a sparsity principle, aiming for the sparsest DAG consistent with the observed distribution. We formulate this as a single differentiable optimization problem. Lastly, we establish theoretical guarantees showing that our method recovers the correct Markov equivalence class under mild assumptions. Unlike prior work, our approach avoids assumptions like linearity or additivity, enabling it to model richer causal dependencies.

In summary, our key contributions are as follows:

1. We start with general binary data, without assuming a particular data generating process, and use the *multivariate Bernoulli* distribution (Definition 1) as a general representation for any such model. We prove that, in this fully general setting, the underlying DAG is *non-identifiable* and characterize which graph–parameter pairs are compatible with the observed distribution (Theorem 1).

2. We formulate learning of the *sparsest* graph and its parameters as a single differentiable program (13). We show that even without assuming faithfulness, any global minimizer yields the sparsest valid DAG (Theorem 3). Moreover, based on our framework, we provide theoretical justification for prior works [13, 5].

3. We conduct experiments demonstrating that our method can capture general causal relationships between discrete variables. Furthermore, we introduce a two-stage approach, BINOTEARS, which reduces computational complexity and outperforms existing baselines.

## 2 Related work

In this section, we survey existing approaches to structure learning and causal discovery, highlighting methods for discrete data, differentiable DAG learning in continuous settings, and recent extensions of continuous-data techniques to discrete domains. We discuss their key ideas, assumptions, and limitations, which motivate the need for a general framework for multivariate discrete data.

**Traditional structure learning for discrete data**    Classical causal-discovery methods for discrete data fall into two main categories. *Constraint-based* algorithms (e.g. PC [41], MMHC [43], Copula-PC [11]) use conditional-independence tests to infer edges. *Score-based* approaches assign each candidate DAG a goodness-of-fit score (e.g. BIC, BDeu [29, 20], generalized scores [22, 27]) and then search for the optimal structure. Greedy strategies such as GES [9] are known to be NP-hard [8, 10] and can get trapped in local optima [26, 2]. Moreover, score-based search traditionally relies on strong parametric assumptions—e.g. additive-noise models [34], hidden compact representations [7], linear effects [27], latent Gaussian variables [1]—which may not hold in practice. A recent exception for general nonparametric models is [3]. A few methods target bivariate causal discovery [45, 28, 6], but they do not fit to multivariate problems. As a result, existing discrete causal-discovery techniques either lack statistical robustness in finite samples or impose restrictive assumptions, motivating the need for scalable, assumption-light methods that handle high-dimensional binary data.

**Continuous DAG learning for continuous data**    Since the introduction of a smooth characterization of acyclicity by Zheng et al. [54], which allows treating the adjacency matrix as a continuous

optimization variable, a large body of work has emerged in differentiable structure learning. Subsequent research has extended this framework in several directions: Nonlinear models [55, 48, 25, 56], optimization theory [44, 32, 14], designing more stable and computationally efficient acyclicity constraints [5, 30, 48, 50, 51], understanding score function properties [31, 15, 40], and incorporating extra side information [4]. Despite this progress, most existing methods are fundamentally designed for *continuous* variables and do not address the specific challenges of discrete data. The common reliance on Gaussianity or continuous-variable assumptions—leading to objectives like mean-squared error or Gaussian log-likelihood—introduces potential biases and inconsistencies when applied to discrete data.

**Continuous DAG learning for discrete data**   Only a handful of recent studies have extended differentiable structure learning to discrete data by imposing specific parametric forms on the conditionals. For example, most works [49, 5, 13] assume a linear SEM in which each discrete node's probability is given by a logistic link on a linear combination of its parents. Such restrictive assumptions can fail to capture the rich, higher-order dependencies present in general discrete distributions. Another line of work in differentiable structure learning studies nonlinear models via neural networks [55, 25, 56], which in principle could accommodate discrete inputs. However, these methods treat inputs as generic real-valued vectors and do not formally address the unique features of discrete data, neither in theory nor in empirical evaluation. A recent contribution by Deng et al. [15] proposes a general differentiable framework for parametric families, but the necessary details to connect generic parametrizations of discrete data to structure learning (in particular, the adjacency matrix), are missing. By contrast, one the of the key contributions in this work is to develop a tractable parametrization via the multivariate Bernoulli distribution that allows explicit representation of the adjacency matrix in a fully differentiable framework without restricting the class of discrete distributions.

## 3   Preliminaries

Let $G = (V, E)$ denote a directed graph on $p$ nodes, with vertex set $V = [p] \coloneqq \{1, \ldots, p\}$ and edge set $E \subset V \times V$, where $(i, j) \in E$ indicates the presence of a directed edge from node $i$ to node $j$. We associate each node $i \in V$ to a random variable $X_i$, and let $X = (X_1, \ldots, X_p)$. In this work we focus on *discrete* random variables. Any variable with $T$ categories can be represented by $(T - 1)$ binary indicators [46]. For instance, if $Z \in \{0, 1, 2\}$, then introduce $Z^k \coloneqq \mathbb{1}\{Z = k\}$, $k = 0, 1, 2$, indicating whether $Z = k$ for $k = 0, 1, 2$. so that $Z$ is replaced by three binary indicators. Hence, without loss of generality, we only consider binary data in the sequel. Extending the results to general discrete data only needs additional notation and does not require new technical ideas.

We review the *multivariate Bernoulli distribution* (MVB) [12], a flexible family that captures *all* joint dependencies among high-dimensional binary variables. Because it assigns a probability to every possible configuration, it allows us to study binary data without imposing additional assumptions.

**Definition 1** (Multivariate Bernoulli distribution [12])**.** *Let $X = (X_1, \ldots, X_p)$ be a vector of possibly correlated Bernoulli random variables with $X_i \in \{0, 1\}$ for $i = 1, \ldots, p$. We say that $X$ follows a multivariate Bernoulli distribution with $\boldsymbol{p} \in \mathbb{R}^{2^p}$, written $X \sim \mathrm{MultiBernoulli}(\boldsymbol{p})$, if*

$$P(X = x) = \prod_{S \subseteq [p]} p(1_S)^{\prod_{j \in S} x_j \ \prod_{j \notin S}(1 - x_j)}. \tag{1}$$

*where $1_S \in \{0, 1\}^p$ for the vector with ones on $S$ and zeros elsewhere and each entry of $\boldsymbol{p} = (p(0, 0, \ldots, 0), \ldots, p(1, \ldots, 1)) \in \mathbb{R}^{2^p}$ is the probability mass of a distinct configuration. Such $\boldsymbol{p}$ is called the general parameter. Let $\mathbf{1}_{2^p} \in \mathbb{R}^{2^p}$ denote the all-ones vector; normalization requires $\mathbf{1}_{2^p}^\top \boldsymbol{p} = 1$.*

Based on (1), it is not clear how to determine the dependence between each variable. For this, define $B^S(x) = \prod_{j \in S} x_j$ for any $S \subseteq [p]$ with convention $B^\emptyset(x) = 1$ . Rewrite (1) in exponential–family form:

$$P(X = x) = \exp\left( \sum_{S \subseteq [p]} f^S B^S(x) \right) \tag{2}$$

where the coefficient $\{f^S\}$ are the natural parameters. The corresponding natural-parameter vector is $\boldsymbol{f} = (f^0, f^1, \ldots, f^p, \ldots, f^{1\ldots p}) \in \mathbb{R}^{2^p}$. Throughout, the superscript set $S$ in $f^S$ is treated as *unordered*; e.g. for $S = \{1, 2\}$ we have $f^{12} = f^{21}$. Because the MVB is an exponential family, there is a one-to-one correspondence between the natural parameters $\boldsymbol{f}$ and the general parameter $\boldsymbol{p}$; explicit conversion formulas are provided in Appendix C.1. Expressing the model as in (2) will later allow us to identify variable dependencies directly from the coefficients $\boldsymbol{f}$ when reconstructing causal graphs. A simple but useful closure property follows immediately.

**Corollary 1.** *If $X \sim \mathrm{MultiBernoulli}(\boldsymbol{p})$, then every marginal distribution and every conditional distribution of $X$ is again multivariate Bernoulli.*

## 4 General discrete data: A nonidentifiable model

We first establish that, for fully general binary data, neither the causal structure nor the associated parameters can be identified from a single observational environment [19, 23]. The intuition is simple: For any fixed topological sort[2] of the variables there exists a unique causal structure and associated parameters that reproduces the observed distribution exactly. Since there are $p!$ distinct topological sorts, this renders the model fundamentally non-identifiable.

### 4.1 Conditional distributions and higher order interaction

First, we provide a useful expression for conditional distributions in this model. Given any permutation $\pi$ on $V$, the joint distribution admits the standard factorization

$$P(X) = \prod_{j=1}^{p} P(X_{\pi(j)} \mid X_{\pi(1),\ldots,\pi(j-1)}).$$

Let us use the joint law $P(X)$ given by the exponential form in (2). Without loss of generality let $\pi = (1, \ldots, p)$ and focus on $j = p$; other choices of $\pi$ and $j$ lead to the same algebra with heavier notation. Then

$$
\begin{aligned}
&\Pr(X_p = 1 \mid X_{-p} = x_{-p}) \\
&= \sigma\left(f^p + f^{1,p}x_1 + \ldots f^{p-1,p}x_{p-1} + f^{12,p}x_1 x_2 + \ldots + f^{1\ldots,p}x_1 \ldots x_{p-1}\right) \\
&= \sigma\left(\sum_{S \subseteq [p-1]} f^{S,p} B^S(x)\right)
\end{aligned}
\tag{3}
$$

where $[p-1] = \{1, \ldots, p-1\}$ and $\sigma(z) = 1/(1 + \exp(-z))$. The full derivation appears in Appendix C.2. Equation (3) highlights that, for general binary data, all higher order interaction terms are present [12, 16]. Omitting these higher-order interactions—e.g., restricting attention to first-order (additive) effects [47, 49]—can misrepresent real data.

### 4.2 Nonidentifiability and Equivalence

In the example above, $X_p$ is the last node in the ordering $\pi$, so any variable in $(X_1, \ldots, X_{p-1})$ may serve as a potential parent of $X_p$. A variable $X_j$ is deemed a parent precisely when some interaction coefficient involving $j$ and $p$ is non-zero; equivalently,

$$X_j \to X_p \Leftrightarrow \exists\, S \subseteq [p-1] \text{ with } j \in S, \text{ such that } f^{S,p} \neq 0 \Leftrightarrow \sum_{S \subseteq [p-1], j \in S} \left(f^{S,p}\right)^2 > 0. \tag{4}$$

All coefficients in (3) can be estimated *simultaneously* via a single logistic regression [24]; deciding whether an edge $X_j \to X_p$ exists thus reduces to checking if the corresponding coefficient set is non-zero. Because (3) involves higher-order interaction terms, we introduce the following notation.

Let $2^{[p]}$ denote the power set of the index set $[p] = \{1, \ldots, p\}$. For any vector $X = (X_1, \ldots, X_p) \in \mathbb{R}^p$ and subset $S \subseteq [p]$, define the *interaction feature* $B^S(X) = \prod_{j \in S} X_j$ with $B^\emptyset(X) = 1$.

---

[2]A *topological sort is a permutation $\pi$ of $V$ such that $X_{\pi(i)} \to X_{\pi(j)} \implies i < j$.*

Collecting all $2^p$ such terms yields the *extended feature map*: $\Phi(X) = \left[B^S(X)\right]_{S \in 2^{[p]}} \in \mathbb{R}^{2^p}$, ordered according to the graded–lexicographic rule described in Appendix C.3. Explicitly,

$$\Phi(X) = \left[1,\ X_1, \ldots, X_p,\ X_1 X_2, \ldots, X_{p-1} X_p,\ X_1 X_2 X_3, \ldots, X_1 \cdots X_p\right]. \tag{5}$$

For example, when $p = 3$ and $X = (X_1, X_2, X_3)$, then the extended feature vector $\Phi(X) = [1, X_1, X_2, X_3, X_1 X_2, X_1 X_3, X_2 X_3, X_1 X_2 X_3] \in \mathbb{R}^8$. When $\Phi$ is applied to data matrix $\mathbf{X} \in \mathbb{R}^{n \times p}$, it operates *row-wise*.

For each $j$ and order $\pi$, define the relevant natural parameters in the vector $\boldsymbol{f}_{\pi,j} \in \mathbb{R}^{2^{j-1}}$ stored in the *same* graded–lexicographic order, but relative to the $(X_{\pi(1)}, \ldots, X_{\pi(j-1)})$. Concretely,

$$\boldsymbol{f}_{\pi,j} = [\underbrace{f_{\pi,j}^{\pi(j)}}_{\text{constant}}, \underbrace{f_{\pi,j}^{\pi(1),\pi(j)}}_{X_{\pi(1)}}, \underbrace{f_{\pi,j}^{\pi(2),\pi(j)}}_{X_{\pi(2)}}, \ldots, \underbrace{f_{\pi,j}^{\pi(j-1),\pi(j)}}_{X_{\pi(j-1)}}, \underbrace{f_{\pi,j}^{\pi(1)\pi(2),\pi(j)}}_{X_{\pi(1)} X_{\pi(2)}}, \ldots, \underbrace{f_{\pi,j}^{\pi(1)\ldots\pi(j-1),\pi(j)}}_{X_{\pi(1)} \ldots X_{\pi(j-1)}}]. \tag{6}$$

Each underbrace indicates the interaction term—either a constant, a single feature, or a product of features from $\{X_{\pi(1)}, \ldots, X_{\pi(j-1)}\}$—that the coefficient corresponds to. Thus, the elements of $\boldsymbol{f}_{\pi,j}$ align exactly with the entries of the vector $\Phi((X_{\pi(1)}, \ldots, X_{\pi(j-1)}))$. The special case used in (3) is recovered by taking $\pi = (1, \ldots, p)$ and $j = p$.

With these definitions in place we can now recover the full causal structure and its parameters. Fix an ordering $\pi$ and consider any $j \in [p]$. By Corollary 1, both the marginal $P(X_{\pi(j)}, X_{\pi(1)}, \ldots, X_{\pi(j-1)})$ and conditional $P(X_{\pi(j)} \mid X_{\pi(1),\ldots,\pi(j-1)})$ remain multivariate Bernoulli. Let $\mathbf{X} = (\mathbf{X}_1, \ldots, \mathbf{X}_p) \in \mathbb{R}^{n \times p}$ be $n$ i.i.d. samples from $X \sim \mathrm{MultiBernoulli}(\boldsymbol{p})$, one sample per row. Hence we can apply the same logic as in Section 4.1:

- Form extended feature vector $\Phi((\mathbf{X}_{\pi(1)}, \ldots, \mathbf{X}_{\pi(j-1)}))$ and run a single logistic regression of $\mathbf{X}_{\pi(j)}$ on these features to recover natural parameters $\boldsymbol{f}_{\pi,j} \in \mathbb{R}^{2^{j-1}}$
- Apply the edge criterion in (4) to $\boldsymbol{f}_{\pi,j}$ to determine the parent set $\mathrm{PA}(\pi(j))$.
- Repeat steps above for $j \in [p]$ to construct the DAG $G_\pi$ and parameter $\boldsymbol{f}_\pi = \{\boldsymbol{f}_{\pi,j}\}_{j=1}^p$.
- Iterate over all permutations $\pi$ to enumerate every graph–parameter pair.

Complete details on such procedure can be found in Appendix C.4.

**Theorem 1.** *Let $\boldsymbol{p} > 0$ and fix any topological sort $\pi$. Consider the population case ($n \to \infty$). Let $(\boldsymbol{f}_\pi, G_\pi)$ be the output of procedure above. If $Y \in \mathbb{R}^p$ are generated by the following structural equations*

$$Y_{\pi(j)} \sim \mathrm{Bernoulli}\left(\sigma\left(\boldsymbol{f}_{\pi,j}^\top \Phi\left((Y_{\pi(1)}, \ldots, Y_{\pi(j-1)})\right)\right)\right) \qquad \forall j = 1, \ldots, p \tag{7}$$

*where $\sigma(z) = 1/(1 + \exp(-z))$, then $Y \sim \mathrm{MultiBernoulli}(\boldsymbol{p})$.*

We work in the population case to eliminate finite sample estimation error. The assumption $\boldsymbol{p} > 0$ is a mild regularity condition routinely adopted in structure learning problems [49, 36, 21] and guarantees that the logistic regressions used in the procedure above always have unique solutions.

**Remark 1.** *Equation (7) expresses a general discrete model as a structural equation model (SEM)[3], a framework widely used for structure learning because it provides a generative description of how each variable arises from its direct causes. A central appeal of SEMs is their universality: Every probability distribution can, in principle, be represented by a suitably chosen SEM (Proposition 7.1 in 35). In practice, however, SEM-based methods almost always impose strong parametric forms—linear, additive, or low-order interactions [47, 49, 18, 52]—which risk ignoring important structure in real data [12, 16]. The problem is particularly acute for discrete data, where higher-order dependencies are easily overlooked, sharply limiting the model's expressive power. By contrast, Theorem 1 shows that general binary distributions necessarily involve higher-order interaction terms. Omitting these terms therefore precludes an exact representation and may lead to incorrect causal conclusions.*

Thus, the Theorem 1 indicates that, for *every* topological order $\pi$ we can construct a distinct SEM that reproduces the same distribution. The model is therefore *non-identifiable*: multiple parameter–graph

---

[3]Formal details can be found in Appendix C.5

pairs $(\boldsymbol{f}_\pi, G_\pi)$ obtained by procedure before give rise to the identical law $\mathrm{MultiBernoulli}(\boldsymbol{p})$. Collect all such pairs into the *equivalence class*

$$\mathcal{E}(\boldsymbol{p}) = \{(\boldsymbol{f}_\pi, G_\pi) : (\boldsymbol{f}_\pi, G_\pi), \forall \pi\}$$

The cardinality of $\mathcal{E}(\boldsymbol{p})$ is at most $p!$, corresponding to the number of permutations of $p$ variables.

**Remark 2.** *It is well-known that causal DAGs are non-identifiable from observational data alone [20, 19, 23]. However, Theorem 1 provides a novel and comprehensive characterization of all DAG-parameter pairs that exactly reproduce a given distribution within the general multivariate Bernoulli framework.*

### 4.3 Minimal equivalence class

Because observational data cannot distinguish among members of $\mathcal{E}(\boldsymbol{p})$, we seek the simplest representation. Specifically, the DAG with the fewest edges. Let $s_G$ denote the number of directed edges in a graph $G$; our objective is to find the pair $(\boldsymbol{f}_\pi, G_\pi) \in \mathcal{E}(\boldsymbol{p})$ with $\min s_{G_\pi}$.

**Definition 2** (Minimality [15])**.** $(\boldsymbol{f}_\pi, G_\pi)$ *is the minimal element in equivalence class* $\mathcal{E}(\boldsymbol{p})$ *if* $s_{G_\pi} \leq s_{G_{\tilde\pi}}, \forall (f_{\tilde\pi}, G_{\tilde\pi}) \in \mathcal{E}(\boldsymbol{p})$. *The set of all the minimal element in equivalence class* $\mathcal{E}(\boldsymbol{p})$ *is referred to as the minimal equivalence class* $\mathcal{E}_{\min}(\boldsymbol{p})$

$$\mathcal{E}_{\min}(\boldsymbol{p}) = \{(\boldsymbol{f}_\pi, G_\pi) : (\boldsymbol{f}_\pi, G_\pi) \text{ is the minimal element}, (\boldsymbol{f}_\pi, G_\pi) \in \mathcal{E}(\boldsymbol{p})\} \tag{8}$$

For the procedure in Section 4.2, if we retains those pairs $(f_\pi, G_\pi)$ with the fewest edges, thereby recovering $\mathcal{E}_{\min}(\boldsymbol{p})$. Minimality is closely related to the *Sparsest Markov Representation* (SMR) assumption [38, 26], which posits that, among all DAGs compatible with a distribution, the sparsest one is unique up to Markov equivalence, i.e., encoding the same conditional independence relationship. SMR is weaker than the well-known *faithfulness* assumption [35]: if a distribution $P$ is faithful to a DAG $G$, then $(G, P)$ satisfies SMR. Hence, targeting the sparsest graph is both principled and practically appealing. Let $\mathcal{M}(G)$ denote the Markov Equivalence class of a DAG $G$, i.e. the set of all DAGs encoding the same conditional-independence relations. Formal definitions of these concepts appears in Appendix C.6.

**Theorem 2.** *For any $\boldsymbol{p} > 0$ and consider the population case $(n \to \infty)$. Under the SMR assumption (or faithfulness assumption), if $(\boldsymbol{f}_{\pi_1}, G_{\pi_1}), (\boldsymbol{f}_{\pi_2}, G_{\pi_2}) \in \mathcal{E}_{\min}(\boldsymbol{p})$, then $\mathcal{M}(G_{\pi_1}) = \mathcal{M}(G_{\pi_2})$*

## 5 Continuous structure learning for general binary data

The procedure in previous section is purely combinatorial and thus fails to scale: It requires fitting $p!$ logistic regressions, so the computational cost grows exponentially with the number of nodes $p$. To overcome this bottleneck, we leverage recent advances in differentiable structure learning [54, 55]. Specifically, we recast the search for the sparsest binary–data DAG as a *single* differentiable program, which can then be tackled by standard gradient-based optimizers.

**Parameterization** Let $\mathbf{X} = (\mathbf{X}_1, \ldots, \mathbf{X}_p)$ be $n$ i.i.d. samples from $X \sim \mathrm{MultiBernoulli}(\boldsymbol{p})$, one sample per row. Define the parameter matrix

$$\mathbf{H}_j = (\underbrace{h^{0,j}}_{\text{constant}}, \underbrace{h^{1,j}, \ldots, h^{p,j}}_{\text{first order}}, \underbrace{h^{12,j}, \ldots, h^{(p-1)p,j}}_{\text{second order}}, \underbrace{\cdots}_{\text{third to } (p-1)\text{-th order}}, \underbrace{h^{123\ldots p,j}}_{p\text{-th order}})^\top \in \mathbb{R}^{2^p} \tag{9}$$

Let $\mathbf{H} = (\mathbf{H}_1, \ldots, \mathbf{H}_p) \in \mathbb{R}^{2^p \times p}$. Here $\mathbf{H}_j$ plays the role of column $j$ in the weighted adjacency matrix. Unlike a linear model, where a single entry $W_{ij}$ signals the edge $X_i \to X_j$ [54], the multiple coefficients in $\mathbf{H}_j$ jointly determine whether such an edge exists.

**Weighted adjacency matrix** Define the induced adjacency matrix

$$[W(\mathbf{H})]_{ij} = \sum_{S \subseteq [p], i \in S} \left(h^{S,j}\right)^2 \tag{10}$$

so $[W(\mathbf{H})]_{ij} > 0$ if and only if some interaction involving $X_i$ contributes to the equation for $X_j$, that is some coefficient $h^{S,j}$ with $i \in S$ is non–zero. Self-loops are forbidden, so we impose

$$h^{S,j} = 0 \quad \text{whenever } j \in S, \qquad \forall j \in [p], \ S \subseteq [p].$$

**Score function**   By Theorem 1 the negative log-likelihood of the multivariate Bernoulli model—-i.e. the logistic or cross entropy loss—provides a suitable score. Applying the extended feature map (5) row-wise to the data matrix $\mathbf{X}$ yields the score function

$$\ell(\mathbf{H}; \mathbf{X}) = \frac{1}{n} \sum_{i=1}^{p} \mathbf{1}_n^\top \left( \log(\mathbf{1}_n + \exp(\Phi(\mathbf{X})\mathbf{H})) - \mathbf{X}_i \circ (\Phi(\mathbf{X})\mathbf{H}) \right)$$

where $\circ$ denotes the Hadamard product and $\exp$ is applied element-wise, see Appendix C.7 for details.

**Regularization**   While $\ell(\mathbf{H}; \mathbf{X})$ identifies the equivalence class $\mathcal{E}(\boldsymbol{p})$, our objective is a *minimal equivalence class* $\mathcal{E}_{\min}(\boldsymbol{p})$. Simply adding a $\ell_0$ [20] penalty would break differentiability, and an $\ell_1$ penalty is both biased and imprecise in edge counting. Instead, we adopt the smooth *quasi minimax-concave penalty (MCP)* [15]:

$$\text{quasi-MCP:} \qquad p_{\lambda,\delta}(t) = \lambda \left[ \left( |t| - \frac{t^2}{2\delta} \right) \mathbb{1}\left( |t| < \delta \right) + \frac{\delta}{2} \mathbb{1}\left( |t| > \delta \right) \right] \tag{11}$$

where $\mathbb{1}(\cdot)$ is the indicator function. $p_{\lambda,\delta}$ is quadratic on $[0, \delta]$ and flat beyond $\delta$, smoothly approximating an $\ell_0$ penalty and thereby encouraging a sparse $W(\mathbf{H})$ without sacrificing differentiability. Let $p_{\lambda,\delta}(W(\mathbf{H})) = \sum_{i \neq j} p_{\lambda,\delta}([W(\mathbf{H})]_{ij})$. Define the regularized score

$$s(\mathbf{H}; \lambda, \delta, \mathbf{X}) = \ell(\mathbf{H}; \mathbf{X}) + p_{\lambda,\delta}(W(\mathbf{H})) \tag{12}$$

**Recovering $\mathcal{E}_{\min}(\boldsymbol{p})$**   We formulate this task as the single continuous optimization problem

$$\min_{\mathbf{H}} \qquad s(\mathbf{H}; \lambda, \delta, \mathbf{X})$$
$$\text{subject to} \qquad h(W(\mathbf{H})) = 0 \tag{13}$$
$$h^{S,j} = 0 \text{ if } j \in S \quad \forall j \in [p], \forall S \subseteq [p]$$

where $h : \mathbb{R}^{p \times p} \to [0, \infty)$ is the differentiable acyclicity constraint [54, 5, 30, 48], satisfying $h(W) = 0$ iff $W$ is a DAG. To study the optimal solutions of (13), define the set of global minimizers

$$\mathcal{O}_{n,\lambda,\delta} = \{(\mathbf{H}^*, G(W(\mathbf{H}^*))) : \mathbf{H}^* \text{ is a minimizer of (13)}\} \tag{14}$$

where $G(W)$ denotes the graph encoded by the adjacency matrix $W$ via (10). Although (13) optimizes only over $\mathbf{H}$, we include the induced graph to match the notation of $\mathcal{E}(\boldsymbol{p})$ and $\mathcal{E}_{\min}(\boldsymbol{p})$.

Ideally, we want the optimal solution set to coincide with the minimal equivalence class, i.e. $\mathcal{O}_{n,\lambda,\delta} = \mathcal{E}_{\min}(\boldsymbol{p})$. However, it is unclear whether suitable values of $(\lambda, \delta)$ exist. The next result shows that, in the population limit, such values can always be chosen. Let $\mathcal{O}_{\infty,\lambda,\delta}$ denote the set of minimizers of (13) when the empirical loss $s(\mathbf{H}; \lambda, \delta, \mathbf{X})$ is replaced by its population counterpart $\mathbb{E}\big[s(\mathbf{H}; \lambda, \delta, \mathbf{X})\big]$.

**Theorem 3.** *For any $\boldsymbol{p} > 0$, there exist $\lambda, \delta > 0$ sufficiently small such that $\mathcal{O}_{\infty,\lambda,\delta} = \mathcal{E}_{\min}(\boldsymbol{p})$. Moreover, under Sparsest Markov Representation (or faithfulness) assumption, for any $(\boldsymbol{f}_{\pi_1}, G_{\pi_1}), (\boldsymbol{f}_{\pi_2}, G_{\pi_2}) \in \mathcal{O}_{\infty,\lambda,\delta}$, it holds $\mathcal{M}(G_{\pi_1}) = \mathcal{M}(G_{\pi_2})$.*

Here, $\mathcal{M}(G)$ is the Markov equivalence class of $G$. The proof adapts Theorem 4 of Deng et al. [15], but requires additional technical work to accommodate the multivariate Bernoulli setting; The theorem implies that, with appropriately small regularization, every global optimum of (13) recovers a sparsest DAG and preserves all conditional-independence relations.

**Connection to prior continuous methods on discrete data.**   Existing empirical studies [13, 5] often generate binary data by passing a *linear* combination of parent variables through a logistic link, then apply continuous-optimization techniques without formal justification. Theorem 3 provides that justification: Such simulations instantiate a special case of our general model. To formalize this setting, we characterize this with the following assumption.

**Assumption A** (First-order logistic model)**.** *There exist a DAG $G^0$ such that $X = (X_1, \ldots, X_p)$ are generated according to the following structure equation model*

$$X_j \mid X_{\text{PA}(j)} \sim \text{Bernoulli}\left( \sigma \left( w_j^\top X + c_j \right) \right) \tag{15}$$

*where $X_{\text{PA}(j)}$ are the parents node of $X_j$ in $G^0$. Here $[w_j]_{[p] \setminus \text{PA}(j)} = 0, [w_j]_{\text{PA}(j)} \neq 0, c_j \in \mathbb{R}$.*

Under Assumption A, only *first-order* interaction terms are present; all higher-order coefficients in (9) vanish. Consequently $\mathbf{H}$ shrinks from $2^p \times p$ to $(p+1) \times p$, and the optimization problem becomes tractable even for moderate $p$. A detailed formulation and corresponding consistency theorem are provided in Appendix C.8.

## 6  Experiments

We solve (13) using either DAGMA [5] or NOTEARS-MLP [55]. Both methods tackle a sequence of unconstrained problems by incorporating the acyclicity constraint as a penalty term with an increasing weight.[4] Our method—denoted BINOTEARS—is compared against several baselines, including DAGMA [5], PC [41], and FGES [37]. Our primary empirical results appear in Figures 1 and 2. We evaluate accuracy using structural Hamming distance (SHD), which measures the discrepancy between the estimated and ground-truth graphs, with lower values indicating better performance. Because the multivariate Bernoulli model is nonidentifiable, we compare completed partially directed acyclic graphs (CPDAGs) of the estimates and the ground truth. Full experimental details are provided in Appendix D.

In Figure 1 (a), we simulate data $\mathbf{X}$ according to the SEM in (7) for small DAG ($d < 10$), including both first-order effects and second-order interactions among parents whenever a node has multiple parents. We observe that BINOTEARS (ours) achieves competitive performance relative to the baselines. In contrast, DAGMA which models only first-order effects, exhibits markedly poor recovery when second-order interactions are present. These results underscore that a purely linear form is insufficient to capture the complex dependencies in general discrete data and can lead to substantial estimation errors whenever higher-order terms play a role. In Figure 1 (b), we repeat the simulation while ablating all first-order effects for nodes with multiple parents, retaining only the highest-order interaction term. Under these conditions, BINOTEARS (ours) consistently attains near-optimal recovery performance across all settings, showing the robustness of our method.

In Figure 2, we consider a larger DAG, where including all possible interaction terms would cause the feature map $\Phi(\mathbf{X})$ to grow exponentially and render the optimization in (13) intractable. To address this, our BINOTEARS works in a two-stage fashion. First, we adapt the original NOTEARS-MLP framework of Zheng et al. [55] to handle discrete data, and use it to learn an initial graph. From this graph we extract a topological ordering $\pi$. In the second (finetuning) stage, we follow the procedure of Section 4.2: we construct only those interaction features consistent with $\pi$, and then fit logistic regressions with first order and second order to recover the final edge structure $G$. BINOTEARS attains performance on par with existing baselines. While all methods degrade as graph size and density increase, our approach exhibits robustness and consistently recovers meaningful structures.

We further evaluate NOTEARS (linear), NOTEARS-MLP, and BINOTEARS (ours) on the real-world dataset of Sachs et al. [39]. The data contain $n = 7{,}466$ measurements of $d = 11$ proteins and phospholipids in human immune cells and are accompanied by a widely used consensus network that serves as a gold standard. Results are summarized in the Table. Further details on the experimental setup and additional results can be found in the Appendices D and E.

| Method | SHD | Num. of Edges |
|---|---|---|
| NOTEARS [54] | 22 | 18 |
| NOTEARS-MLP [55] | 16 | 13 |
| BINOTEARS (Ours) | 13 | 15 |

## 7  Conclusion

We have introduced a fully general differentiable framework for structure learning in binary data, based on the multivariate Bernoulli distribution. Our key contributions include (i) a complete characterization of non-identifiability in the general discrete setting and a constructive procedure

---

[4]Concretely, we (i) augment DAGMA with higher-order terms on **small** graphs to pursue exact recovery, and (ii) employ NOTEARS-MLP to obtain a topological order on **large** graphs and subsequently fit logistic regressions for edge selection. We use these two pathways because, in our experiments, they strike different speed–accuracy trade-offs. See Appendix D.2 for implementation details.

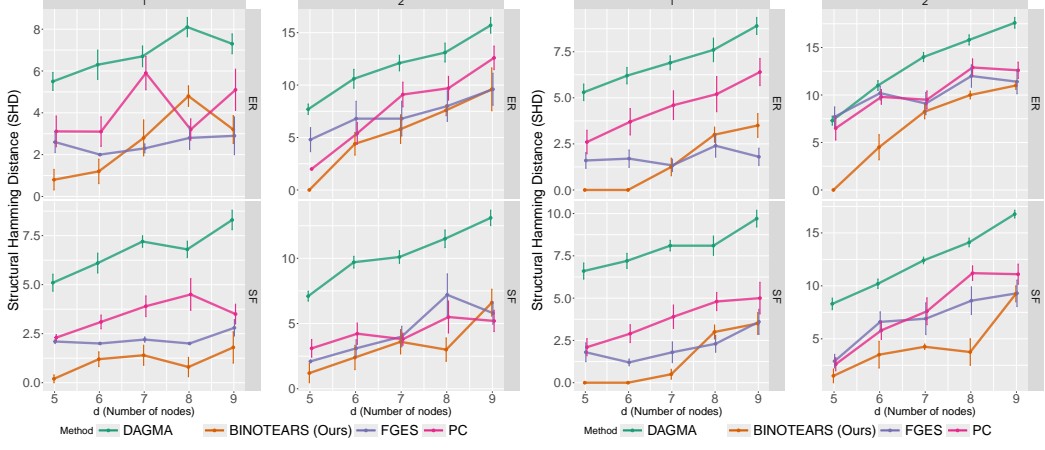

(a) First order and second order interaction are involved     (b) Only highest order interaction is involved

Figure 1: Results in terms of SHD between MECs of estimated graph and ground truth. Lower is better. Column: $k = \{1, 2\}$. Row: random graph types. $\{ER, SF\}$-$k = \{Scale\text{-}Free, Erdős–Rényi\}$ graphs with $k \cdot d$ expected edges. Here $p = \{5, 6, 7, 8, 9\}$. Error bars denote the standard error computed over 10 replications.

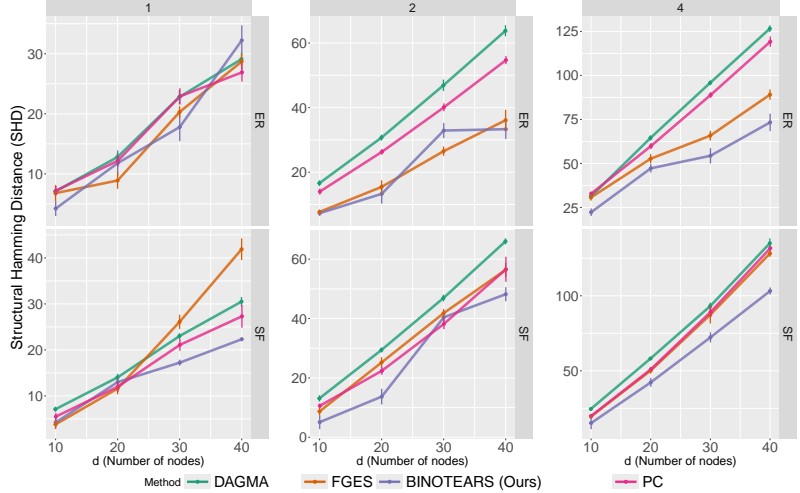

Figure 2: Results in terms of SHD between MECs of estimated graph and ground truth. Lower is better. Column: $k = \{1, 2, 4\}$. Row: random graph types. $\{ER, SF\}$-$k = \{Scale\text{-}Free, Erdős–Rényi\}$ graphs with $kd$ expected edges. Here $p = \{10, 20, 30, 40\}$. BɪNOTEARS is our two-stage approach.

to recover the minimal equivalence class (Theorems 1–2), (ii) a fully differentiable constrained programming that provably recovers a sparsest DAG up to Markov equivalence (Theorem 3), and (iii) empirical validation on synthetic graphs of varying size, demonstrating that our method captures complex higher-order dependencies where existing approaches fail.

To scale beyond small graphs, we further propose a practical two-stage heuristic, BɪNOTEARS, which first learns a coarse structure via NOTEARS-MLP adapted to discrete data and then refines edge estimates by fitting logistic regressions along the inferred topological order. While this approach yields strong numerical performance on larger networks, its theoretical properties remain unexplored. In future work, we aim to develop principled, scalable algorithms for discrete structure learning that retain both computational tractability and rigorous identifiability guarantees in regimes with thousands of nodes.

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

# SUPPLEMENTARY MATERIAL
## Differentiable Structure Learning for General Binary Data

## A  Preliminary Technical Results

In this appendix, we present several key technical results that are essential to our proofs.

**Lemma 1.** *Suppose $X \sim \mathrm{MultiBernoulli}(\boldsymbol{p})$ with $\boldsymbol{p} > 0$. Then for every subset $S \subseteq [p]$, the marginal vector $X_S$ satisfies*

$$X_S \sim \mathrm{MultiBernoulli}(\boldsymbol{p}_S), \tag{16}$$

*for certain $\boldsymbol{p}_S \in \mathbb{R}^{|S|}$ where $\boldsymbol{p}_S > 0$.*

The marginal vector $X_S$ remains multivariate Bernoulli with natural parameter $\boldsymbol{p}_S$, and positivity of $\boldsymbol{p}$ indicates the positivity of $\boldsymbol{p}_S$.

**Lemma 2.** *Suppose $X \sim \mathrm{MultiBernoulli}(\boldsymbol{p})$ in the natural parameterization, or equivalently $X \sim \mathrm{MultiBernoulli}(\boldsymbol{f})$ in the general parameterization. Then*

$$\boldsymbol{p} > 0 \quad \Longleftrightarrow \quad |\boldsymbol{f}| < \infty. \tag{17}$$

The general parameter vector $\boldsymbol{p}$ is strictly positive if and only if its corresponding natural parameter $\boldsymbol{f}$ is strictly finite.

**Lemma 3.** *Let $\boldsymbol{p} > 0$ and suppose $X \sim \mathrm{MultiBernoulli}(\boldsymbol{p})$. Fix any topological order $\pi$ and index $j \in [p]$. Define the population negative log-likelihood*

$$\ell(w) = \mathbb{E}\Big[\log\big(1 + \exp\big(w^\top \Phi(X_{\pi(1)}, \dots, X_{\pi(j-1)})\big)\big) - X_{\pi(j)}\, w^\top \Phi(X_{\pi(1)}, \dots, X_{\pi(j-1)})\Big] \tag{18}$$

*where $w \in \mathbb{R}^{2^{j-1}}$. Then $\ell(w)$ is strictly convex and therefore admits a* unique *minimizer $w_{\pi,j}^*$.*

This optimization problem is equivalent to fitting, in the population limit, a logistic regression of $X_{\pi(j)}$ on $\Phi\big((X_{\pi(1)}, \dots, X_{\pi(j-1)})\big)$. Because $\boldsymbol{f}_{\pi,j}$, as defined in Section 4.2, is one solution to the optimization above, and we show optimal solution is unique, then $w_{\pi,j}^* = \boldsymbol{f}_{\pi,j}$. Consequently, logistic regression perfectly recovers the true natural parameter.

**Corollary 2.** *Let $f : \mathbb{R}^d \to \mathbb{R}$ be differentiable and $m$-strongly convex, i.e.*

$$f(y) \geq f(x) + \nabla f(x)^\top (y - x) + \frac{m}{2}\|y - x\|^2 \qquad \forall x, y \in \mathbb{R}^d \tag{19}$$

*Denote by $x^*$ the unique minimizer $f$, so $f(x^*) = \min_x f(x)$. Then for any constant $c \geq f(x^*)$ the subset*

$$L_c = \{x \in \mathbb{R}^p : f(x) \leq c\} \tag{20}$$

*is bounded.*

Corollary 2 asserts that any strongly convex function possesses bounded level sets, which is used in the proof of Theorem 3.

## B  Proofs

In this appendix, we provide detailed proofs of the main results.

### B.1  Proof of Lemma 1

From the proof of Corollary 1, we know that $X_S \sim MultiBernoulli(\boldsymbol{p}_S)$. Moreover, because $\boldsymbol{p} > 0$, then

$$P(X_S = x_s) = \sum_{x_{[p]\backslash S} \in \{0,1\}^{p-|S|}} P(X_S = x_S, X_{[p]\backslash S} = x_{[p]\backslash S}) > 0 \tag{21}$$

Therefore, for any $x_S$, $P(X_S = x_s) > 0$. So, $\boldsymbol{p}_S > 0$.

## B.2 Proof of Lemma 2

**Sufficiency** By (67), each natural-parameter probability satisfies

$$\exp(f^{j_1 j_2 \cdots j_r}) \tag{22}$$

$$= \frac{\prod p(\text{even \# zeros among } j_1, j_2, \ldots, j_r \text{ components and other components are all zero})}{\prod p(\text{odd \# zeros among } j_1, j_2, \ldots, j_r \text{ components and other components are all zero})}, \tag{23}$$

and since $\boldsymbol{p} > 0$, every term $\exp(f^{j_1 \cdots j_r})$ is strictly positive and finite. Hence

$$0 < \exp(f^{j_1 \cdots j_r}) < \infty \quad \Longleftrightarrow \quad |f^{j_1 \cdots j_r}| < \infty. \tag{24}$$

**Necessity** Note that

$$S^{j_1 j_2 \cdots j_r} = \sum_{1 \le s \le r} f^{j_s} + \sum_{1 \le s < t \le r} f^{j_s j_t} + \ldots + f^{j_1 j_2 \cdots j_r} \tag{25}$$

If $|\boldsymbol{f}| < \infty$, then $|S^{j_1 \cdots j_r}| < \infty$, so

$$0 < \exp(S^{j_1 j_2 \cdots j_r}) < \infty \tag{26}$$

The joint probability of observing ones in positions $j_1, \ldots, j_r$ is

$$P(j_1, j_2, \ldots, j_r \text{ positions are one, others are zero})$$
$$= \frac{\exp(S^{j_1 j_2 \cdots j_r})}{\exp(b(\boldsymbol{f}))}.$$
$$= \frac{\exp(S^{j_1 j_2 \cdots j_r})}{\sum_{r=1}^{K} \left[ 1 + \left( \sum_{1 \le j_1 < j_2 < \ldots < j_r \le K} \exp[S^{j_1 j_2 \cdots j_r}] \right) \right]} \tag{27}$$

which lies in the interval $(0, 1)$:

$$0 < P(j_1, j_2, \ldots, j_r \text{ positions are one, others are zero}) < 1 \tag{28}$$

## B.3 Proof of Lemma 3

Fix a topological order $\pi$ and an index $j \in [p]$. To simplify notation, set

$$Y = X_{\pi(j)} \in \{0, 1\}, \qquad Z = \Phi\big((X_{\pi(1)}, \ldots, X_{\pi(j-1)})\big) \in \{0, 1\}^{2^{j-1}}, \tag{29}$$

and let $w \in \mathbb{R}^{2^{j-1}}$ be the parameter vector. The population objective is

$$\ell(w) = \mathbb{E}\Big[\log\big(1 + \exp(w^\top Z)\big) - Y(w^\top Z)\Big]. \tag{30}$$

A straightforward calculation shows

$$\nabla^2 \ell(w) = \mathbb{E}\Big[\sigma(w^\top Z)\big(1 - \sigma(w^\top Z)\big) Z Z^\top\Big], \quad \sigma(c) = \frac{1}{1 + e^{-c}}. \tag{31}$$

Since $Z \in \{0, 1\}^{2^{j-1}}$, every inner product $w^\top Z$ is finite and thus $\sigma(w^\top Z)(1 - \sigma(w^\top Z)) > 0$.
Define

$$m(w) = \min_{z \in \{0,1\}^{2^{j-1}}} \sigma(w^\top z)\big(1 - \sigma(w^\top z)\big) > 0. \tag{32}$$

The marginal distribution of $(X_{\pi(1)}, \ldots, X_{\pi(j-1)})$ is still a multivariate Bernoulli distribution, with natural parameters $\boldsymbol{p}_{\pi,j-1}$, i.e., $(X_{\pi(1)}, \ldots, X_{\pi(j-1)}) \sim MultiBernoulli(\boldsymbol{p}_{\pi,j-1})$. By lemma 1, $q_{\min,j-1} = \min_{i \in [2^{j-1}]} [\boldsymbol{p}_{\pi,j-1}]_i > 0$. Hence for any nonzero $\beta \in \mathbb{R}^{2^{j-1}}$,

$$\beta^\top \nabla^2 \ell(w)\beta = \sum_{z \in \{0,1\}^{2^{j-1}}} p(z)\,\sigma(w^\top z)(1 - \sigma(w^\top z))\,(\beta^\top z)^2$$

$$\ge q_{\min,j-1}\,m(w) \sum_{z \in \{0,1\}^{2^{j-1}}} (\beta^\top z)^2 > 0, \tag{33}$$

because the sum is over all $z \in \{0,1\}^{2^{j-1}}$. So $(\beta^\top z)^2 > 0$ for at least one $z$ when $\beta \neq 0$.

Since the Hessian is everywhere positive definite, $\ell(w)$ is strictly convex and therefore has a unique minimizer.

## B.4  Proof of Theorem 1

Let $\pi$ be any permutation of $\{1, \ldots, p\}$. Then the joint distribution of $X$ can be written as

$$P(X) = \prod_{j=1}^{p} P(X_{\pi(j)} \mid X_{\pi(1),\ldots,\pi(j-1)}) \tag{34}$$

In the population limit ($n \to \infty$), each conditional law has the Bernoulli form

$$P(X_{\pi(j)} \mid X_{\pi(1)}, \ldots, X_{\pi(j-1)}) = q^{X_{\pi(j)}} (1-q)^{1-X_{\pi(j)}} \tag{35}$$

where

$$q = \sigma\left( \boldsymbol{f}_{\pi,j}^\top \Phi\left( (X_{\pi(1)}, \ldots, X_{\pi(j-1)}) \right) \right) \tag{36}$$

Lemma 3 guarantees that logistic regression uniquely recovers $\boldsymbol{f}_{\pi,j}$. Moreover, the structural equation model

$$Y_{\pi(j)} \sim \text{Bernoulli}\left( \sigma\left( \boldsymbol{f}_{\pi,j}^\top \Phi\left( (Y_{\pi(1)}, \ldots, Y_{\pi(j-1)}) \right) \right) \right) \tag{37}$$

It induces exactly the same conditionals as $X$. i.e.,

$$Y_{\pi(j)} \mid Y_{\pi(1)}, \ldots, Y_{\pi(j-1)} \stackrel{d}{=} X_{\pi(j)} \mid X_{\pi(1)}, \ldots, X_{\pi(j-1)}, \tag{38}$$

Hence

$$Y \sim \text{MultiBernoulli}(\boldsymbol{p}), \tag{39}$$

establishing the desired result.

## B.5  Proof of Theorem 2

By Theorem 1, for every $(\boldsymbol{f}_\pi, G_\pi) \in \mathcal{E}_{\min}(\boldsymbol{p})$, the vector $\boldsymbol{f}_\pi$ defines, via the structural equation model (7), a distribution

$$X \sim \text{MultiBernoulli}(\boldsymbol{p}) \tag{40}$$

that is Markov with respect to $G_\pi$. By definition of $\mathcal{E}_{\min}(\boldsymbol{p})$, each $G_\pi$ is a sparsest graph in the equivalence class $\mathcal{E}(\boldsymbol{p})$. Since all sparsest Markov representations lie in the same Markov equivalence class, it follows that for any two pairs

$$(\boldsymbol{f}_{\pi_1}, G_{\pi_1}), \ (\boldsymbol{f}_{\pi_2}, G_{\pi_2}) \ \in \ \mathcal{E}_{\min}(\boldsymbol{p}), \tag{41}$$

their Markov classes coincide:

$$\mathcal{M}(G_{\pi_1}) \ = \ \mathcal{M}(G_{\pi_2}). \tag{42}$$

## B.6  Proof of Theorem 3

The proof relies on Theorems 4 and 5 of Deng et al. [15]. We verify Assumptions A and B from that work.

**Assumption A (1)**  This requires the equivalence class to be finite. Since

$$|\mathcal{E}_{\min}(\boldsymbol{p})| \leq p! \tag{43}$$

the condition holds.

**Assumption A (2)** This requires the weighted adjacency matrix $W(\mathbf{H})$ to be $L$-Lipschitz. Recall that in (10),

$$[W(\mathbf{H})]_{ij} = \sum_{S \subseteq [p],\, i \in S} \left( h^{S,j} \right)^2. \tag{44}$$

For simplicity we instead use the equivalent form

$$[W(\mathbf{H})]_{ij} = \sum_{S \subseteq [p],\, i \in S} \left| h^{S,j} \right|. \tag{45}$$

The reason of equivalence is that they encode the same nonzero pattern of $[W(\mathbf{H})]_{ij}$. Let $\mathbf{H}_1$ and $\mathbf{H}_2$ be two parameter values. We show there exists $L$ such that

$$\|W(\mathbf{H}_1) - W(\mathbf{H}_2)\|_2 \le L \|\mathbf{H}_1 - \mathbf{H}_2\|_2. \tag{46}$$

First,

$$\|W(\mathbf{H}_1) - W(\mathbf{H}_2)\|_2 = \sqrt{\sum_j \sum_i \left( \sum_{S \subseteq [p],\, i \in S} \left| h_1^{S,j} \right| - \sum_{S \subseteq [p],\, i \in S} \left| h_2^{S,j} \right| \right)^2}. \tag{47}$$

Meanwhile,

$$\|\mathbf{H}_1 - \mathbf{H}_2\|_2 = \sqrt{\sum_j \sum_{S \subseteq [p]} \left( h_1^{S,j} - h_2^{S,j} \right)^2}. \tag{48}$$

By Cauchy–Schwarz,

$$
\begin{aligned}
\left( \sum_{S \subseteq [p],\, i \in S} \left| h_1^{S,j} \right| - \sum_{S \subseteq [p],\, i \in S} \left| h_2^{S,j} \right| \right)^2 &= \left( \sum_{S \subseteq [p],\, i \in S} \left( \left| h_1^{S,j} \right| - \left| h_2^{S,j} \right| \right) \right)^2 \\
&\le |S \subseteq [p],\, i \in S| \sum_{S \subseteq [p],\, i \in S} \left( \left| h_1^{S,j} \right| - \left| h_2^{S,j} \right| \right)^2 \\
&\le 2^{p-1} \sum_{S \subseteq [p],\, i \in S} \left( h_1^{S,j} - h_2^{S,j} \right)^2 \\
&\le 2^{p-1} \sum_{S \subseteq [p]} \left( h_1^{S,j} - h_2^{S,j} \right)^2
\end{aligned}
\tag{49}
$$

Hence

$$
\begin{aligned}
\|W(\mathbf{H}_1) - W(\mathbf{H}_2)\|_2 &= \sqrt{\sum_j \sum_i \left( \sum_{S \subseteq [p],\, i \in S} \left| h_1^{S,j} \right| - \sum_{S \subseteq [p],\, i \in S} \left| h_2^{S,j} \right| \right)^2} \\
&\le \sqrt{\sum_j \sum_i 2^{p-1} \sum_{S \subseteq [p]} \left( h_1^{S,j} - h_2^{S,j} \right)^2} \\
&\le \sqrt{2^{p-1} p \sum_j \sum_{S \subseteq [p]} \left( h_1^{S,j} - h_2^{S,j} \right)^2} \\
&= \sqrt{p}\, 2^{(p-1)/2} \sqrt{\sum_j \sum_{S \subseteq [p]} \left( h_1^{S,j} - h_2^{S,j} \right)^2} \\
&= \sqrt{p}\, 2^{(p-1)/2} \|\mathbf{H}_1 - \mathbf{H}_2\|_2
\end{aligned}
\tag{50}
$$

Thus one may take $L = \sqrt{p}\, 2^{(p-1)/2}$.

**Assumption B** This requires $\mathbb{E}[s(\mathbf{H}; \mathbf{X})]$ to have bounded level sets. From Lemma 3 we know that under $\boldsymbol{p} > 0$, each population logistic loss is strongly convex. Since $\mathbb{E}[s(\mathbf{H}; \mathbf{X})]$ is a sum of $p$ such strongly convex terms, it is itself strongly convex. By Corollary 2, any strongly convex function has bounded sublevel sets.

## B.7 Proof of Theorem 4

Let $\pi^*$ be the topological ordering of $G$ guaranteed by Assumption A. Then

$$P(X) = \prod_{j=1}^{p} P(X_{\pi^*(j)} \mid X_{\pi^*(1),\ldots,\pi^*(j-1)}) \tag{51}$$

Write $S_{\pi,j} = \{\pi^*(1), \ldots, \pi^*(j-1)\}$. Under the linear SEM assumption,

$$X_{\pi^*(j)} \mid X_{S_{\pi,j}} = x_{S_{\pi,j}} \sim \text{Bernoulli}(\sigma([w_{\pi^*(j)}]_{S_{\pi,j}}^{\top} X_{S_{\pi,j}} + c_{\pi^*(j)})) \tag{52}$$

where $w_{\pi^*(j)}$ and $c_{\pi^*(j)}$ are finite. Hence for any $x_{S_{\pi,j}}$,

$$0 < \sigma\big(w_{\pi^*(j), S_{\pi,j}}^{\top} x_{S_{\pi,j}} + c_{\pi^*(j)}\big) < 1, \tag{53}$$

so both

$$P\big(X_{\pi^*(j)} = 1 \mid X_{S_{\pi,j}} = x_{S_{\pi,j}}\big) \quad \text{and} \quad P\big(X_{\pi^*(j)} = 0 \mid X_{S_{\pi,j}} = x_{S_{\pi,j}}\big) \tag{54}$$

lie in $(0, 1)$, implying $\boldsymbol{p} > 0$.

Although every $(\boldsymbol{f}_{\pi}, G_{\pi}) \in \mathcal{E}(\boldsymbol{p})$ generates $X \sim \text{MultiBernoulli}(\boldsymbol{p})$ via (7), the linear optimization (99) only admits first-order models. Thus any $(\boldsymbol{f}_{\pi}, G_{\pi})$ with higher-order terms is misspecified. Define

$$\mathcal{E}^{linear}(\boldsymbol{p}) = \{(\boldsymbol{f}_{\pi}, G_{\pi}) : (\boldsymbol{f}_{\pi}, G_{\pi}) \text{ where } \boldsymbol{f}_{\pi} \text{ only has first order term}, \forall \pi\} \tag{55}$$

By Assumption A, $(W^0, G^0) \in \mathcal{E}^{\text{linear}}(\boldsymbol{p})$. Let

$$\mathcal{E}_{\min}^{linear}(\boldsymbol{p}) = \{(\boldsymbol{f}_{\pi}, G_{\pi}) : (\boldsymbol{f}_{\pi}, G_{\pi}) \text{ is the minimal element}, (\boldsymbol{f}_{\pi}, G_{\pi}) \in \mathcal{E}^{linear}(\boldsymbol{p})\} \tag{56}$$

If $(W^0, G^0)$ is not already minimal, we simply replace it by the sparsest element in $\mathcal{E}^{\text{linear}}(\boldsymbol{p})$, so that $(W^0, G^0) \in \mathcal{E}_{\min}^{\text{linear}}(\boldsymbol{p})$.

In the proof of Theorem 3, we verified that our model meets Assumptions A and B of Deng et al. [15]. Therefore, by Theorem 4 of Deng et al. [15], we have

$$\mathcal{O}_{\infty,\lambda,\delta}^{\text{linear}} = \mathcal{E}_{\min}^{\text{linear}}(\boldsymbol{p}), \quad \text{and hence} \quad (W^0, G^0) \in \mathcal{O}_{\infty,\lambda,\delta}^{\text{linear}}. \tag{57}$$

## B.8 Proof of Corollary 1

Since $X \sim MultiBernoulli(\boldsymbol{p})$ and $X = (X_1, \ldots, X_p)$. Then, $\forall S \subseteq [p]$, the range of $X_S \in \{0,1\}^{|S|}$. The corresponding density mass

$$P(X_S = x_s) = \sum_{x_{[p]\setminus S} \in \{0,1\}^{p-|S|}} P(X_S = x_S, X_{[p]\setminus S} = x_{[p]\setminus S}) \tag{58}$$

In such way, we could enumerate all the value for the $x_s \in \{0,1\}^{|S|}$, and put them together to get the natural parameter for $X_S$, i.e., $\boldsymbol{p}_S$. As a consequence, $X_S \sim MultiBernoulli(\boldsymbol{p}_S)$.

For any $j \in [p]$, and any $S \in [p]\setminus j$, since $X_j \in \{0,1\}$, so the conditional distribution $P(X_j \mid X_S)$ is Bernoulli distribution, and the probability

$$\begin{aligned} P(X_j = x_j \mid X_S = x_S) &= \frac{P(X_j = x_j, X_S = x_S)}{P(X_S = x_S)} \\ &= \frac{P(X_j = x_j, X_S = x_S)}{P(X_j = 1, X_S = x_S) + P(X_j = 0, X_S = x_S)} \end{aligned} \tag{59}$$

### B.9 Proof of Corollary 2

Since $f$ is strongly convex,

$$f(y) \geq f(x) + \nabla f(x)^\top (y - x) + \frac{m}{2} \|y - x\|^2 \qquad \forall x, y \in \mathbb{R}^d \tag{60}$$

Take $x = x^*$, then

$$f(y) \geq f(x^*) + \frac{m}{2} \|y - x^*\|^2 \qquad \forall x, y \in \mathbb{R}^d \tag{61}$$

Rearranging,

$$f(y) \leq c \Rightarrow \frac{m}{2} \|y - x^*\|^2 \leq c - f(x^*) \Rightarrow \|y - x^*\| \leq \sqrt{\frac{2(c - f(x^*))}{m}} \tag{62}$$

Thus, $L_c \subseteq \left\{ y : \|y - x^*\| \leq \sqrt{\frac{2(c-f(x^*))}{m}} \right\}$, a bounded ball.

## C Supplementary Technical Details and Examples

In this appendix, we collect additionsl technical derivations, algorithms, and definitions that support our main theorems.

- In C.1, we give the explicit parameter transformation between the general parameter $p$ and the natural parameter $f$ of the multivariate Bernoulli model [12];
- In C.2, we derive the logistic form of the conditional distributions in equation (3)
- In C.3, we formalize the graded–lexicographic ordering used to index the interaction features.
- In C.4, we present the population-level recovery algorithms for each topological order $\pi$.
- In C.5, we define the structural equation model framework underlying Theorem 1.
- In C.6, we review faithfulness, Markov equivalence, and the Sparsest Markov Representation necessary for Theorems 2 and 3
- In C.7, we provide the derivation of our score function $s(\mathbf{H}; \mathbf{X})$ (negative log-likelihood function).
- In C.8, we provide theoretical justification for the previous work [13, 5] with our general framework.

### C.1 Parameter Transformation in the Multivariate Bernoulli Distribution

All the material in this subsection can be found in [12]. We include here for the completeness.

The multivariate Bernoulli distribution has two different parameterization, one is using general parameter $p$ and another one is using natural parameter $f$.

The density is expressed by general parameter $p$.

$$
\begin{aligned}
P(Y_1 = y_1, Y_2 = y_2, \ldots, Y_K = y_K) &= p(y_1, y_2, \ldots, y_K) \\
&= p(0, 0, \ldots, 0)^{\Pi_{j=1}^K (1-y_j)} \\
&\quad \times p(1, 0, \ldots, 0)^{\left[ y_1 \prod_{j=2}^K (1-y_j) \right]} \\
&\quad \times p(0, 1, \ldots, 0)^{\left[ (1-y_1) y_2 \prod_{j=3}^K (1-y_j) \right]} \ldots \\
&\quad \times p(1, 1, \ldots, 1)^{\Pi_{j=1}^K y_j},
\end{aligned} \tag{63}
$$

The density is expressed by natural parameter $f$.

$$P(Y_1 = y_1, Y_2 = y_2, \ldots, Y_K = y_K) = \exp \left( f^0 + \sum_{r=1}^{p} \left( \sum_{1 \leq j_1 < j_2 < \ldots < j_r \leq p} f^{j_1 j_2 \ldots j_r} B^{j_1 j_2 \ldots j_r}(y) \right) \right) \tag{64}$$

To simplify the notation, we could define the quantity $S$ to be

$$S^{j_1 j_2 \cdots j_r} = \sum_{1 \le s \le r} f^{j_s} + \sum_{1 \le s < t \le r} f^{j_s j_t} + \ldots + f^{j_1 j_2 \cdots j_r} \tag{65}$$

and also define the interaction function $B$

$$B^{j_1 j_2 \cdots j_r}(y) = y_{j_1} y_{j_2} \cdots y_{j_r} \tag{66}$$

The following lemma shows the one-to-one mapping between general parameter $p$ and natural parameter $f$.

**Lemma 4** (Parameter transformation). *For the multivariate Bernoulli model, the general parameters and natural parameters have the following relationship.*

$$\exp(f^{j_1 j_2 \cdots j_r}) \tag{67}$$

$$= \frac{\prod p(\text{even \# zeros among } j_1, j_2, \ldots, j_r \text{ components and other components are all zero})}{\prod p(\text{odd \# zeros among } j_1, j_2, \ldots, j_r \text{ components and other components are all zero})}, \tag{68}$$

*where # refers to the number of zeros among the superscript $y_{j_1} \cdots y_{j_r}$ of $f$. In addition,*

$$\exp(S^{j_1 j_2 \cdots j_r}) = \frac{p(j_1, j_2, \ldots, j_r \text{ positions are one, others are zero})}{p(0, 0, \ldots, 0)} \tag{69}$$

*and conversely the general parameters can be represented by the natural parameters*

$$p(j_1, j_2, \ldots, j_r \text{ positions are one, others are zero}) = \frac{\exp(S^{j_1 j_2 \cdots j_r})}{\exp(b(f))}. \tag{70}$$

*where*

$$b(f) = \log \sum_{r=1}^{K} \left[ 1 + \left( \sum_{1 \le j_1 < j_2 < \ldots < j_r \le K} \exp[S^{j_1 j_2 \cdots j_r}] \right) \right] \tag{71}$$

## C.2 Conditional distribution of multivariate Bernoulli distribution

In this part, we derive the conditional distribution of multivariate Bernoulli distribution. Especially,

$$P(X_p = 1 \mid X_1 = x_1, \ldots, X_{p-1} = x_{p-1})$$

$$= \sigma \left( \sum_{r=1}^{p} \left( \sum_{1 \le j_1 < j_2 < \ldots < j_r = p \le p} f^{j_1 \ldots j_{r-1} p} x_{j_1} \ldots x_{j_{r_1}} x_p \right) \right) \tag{72}$$

$$= \sigma \left( f^p + f^{1p} x_1 + f^{2p} x_2 \ldots f^{p-1,p} f_{p-1} + f^{12p} x_1 x_2 + \ldots + f^{1 \ldots p} x_1 \ldots x_{p-1} \right)$$

It is known the multivariate Bernoulli distribution in exponential form can be written as

$$P(X_1 = x_1, \ldots, X_p = x_p) = \exp \left( f^0 + \sum_{r=1}^{p} \left( \sum_{1 \le j_1 < j_2 < \ldots < j_r \le p} f^{j_1 j_2 \cdots j_r} B^{j_1 j_2 \cdots j_r}(x) \right) \right) \tag{73}$$

Then,

$$P(X_p = 1 \mid X_{-p} = x_{-p}) = \frac{P(X_p = 1, X_{-p} = x_{-p})}{P(X_{-p} = x_{-p})}$$

$$= \frac{P(X_p = 1, X_{-p} = x_{-p})}{P(X_p = 1, X_{-p} = x_{-p}) + P(X_p = 0, X_{-p} = x_{-p})} \tag{74}$$

$$= \frac{1}{1 + \frac{P(X_p = 0, X_{-p} = x_{-p})}{P(X_p = 1, X_{-p} = x_{-p})}}$$

where $X_{-p} = (X_1, \ldots, X_{p-1})$.

$$P(X_1 = x_1, \ldots, X_p = x_p)$$

$$= \exp\left(f^0 + \sum_{r=1}^p \left(\sum_{1 \le j_1 < j_2 < \ldots < j_r \le p} f^{j_1 j_2 \ldots j_r} B^{j_1 j_2 \ldots j_r}(x)\right)\right)$$

$$= \exp\left(f^0 + \sum_{r=1}^p \left(\sum_{\substack{1 \le j_1 < j_2 < \ldots < j_r \le p \\ p \in \{j_1, \ldots, j_r\}}} f^{j_1 j_2 \ldots j_r} B^{j_1 j_2 \ldots j_r}(x) + \sum_{\substack{1 \le j_1 < j_2 < \ldots < j_r \le p \\ p \notin \{j_1, \ldots, j_r\}}} f^{j_1 j_2 \ldots j_r} B^{j_1 j_2 \ldots j_r}(x)\right)\right)$$

$$(75)$$

Then,

$$P(X_1 = x_1, \ldots, X_p = 1)$$

$$= \exp\left(f^0 + \sum_{r=1}^p \left(\sum_{\substack{1 \le j_1 < j_2 < \ldots < j_r \le p \\ p \in \{j_1, \ldots, j_r\}}} f^{j_1 j_2 \ldots j_r} B^{j_1 j_2 \ldots j_r}((x_{-p}, 1)) + \sum_{\substack{1 \le j_1 < j_2 < \ldots < j_r \le p \\ p \notin \{j_1, \ldots, j_r\}}} f^{j_1 j_2 \ldots j_r} B^{j_1 j_2 \ldots j_r}(x)\right)\right)$$

$$(76)$$

$$P(X_1 = x_1, \ldots, X_p = 0) = \exp\left(f^0 + \sum_{r=1}^p \left(\sum_{\substack{1 \le j_1 < j_2 < \ldots < j_r \le p \\ p \notin \{j_1, \ldots, j_r\}}} f^{j_1 j_2 \ldots j_r} B^{j_1 j_2 \ldots j_r}(x)\right)\right) \quad (77)$$

Finally, put them together,

$$P(X_p = 1 \mid X_{-p} = x_{-p})$$

$$= \frac{1}{1 + \frac{P(X_p=0, X_{-p}=x_{-p})}{P(X_p=1, X_{-p}=x_{-p})}}$$

$$= \frac{1}{1 + \exp\left(-\sum_{r=1}^p \left(\sum_{\substack{1 \le j_1 < j_2 < \ldots < j_r \le p \\ p \in \{j_1, \ldots, j_r\}}} f^{j_1 j_2 \ldots j_r} B^{j_1 j_2 \ldots j_r}((x_{-p}, 1))\right)\right)} \quad (78)$$

$$= \sigma\left(f^p + f^{1p}x_1 + f^{2p}x_2 \ldots f^{p-1,p}f_{p-1} + f^{12p}x_1 x_2 + \ldots + f^{1 \ldots p}x_1 \ldots x_{p-1}\right)$$

where $\sigma(x) = \frac{1}{1+\exp(-x)}$.

### C.3 Graded-lexicographic order

To index the $2^p$ interaction features and corresponding parameters in a consistent way, we use the *graded–lexicographic* order on subsets of $[p]$, where for any finite set $S$, $|S|$ denotes its cardinality (the number of elements in $S$).

**Definition 3** (Graded–lexicographic order). *Let $\pi$ be a permutation of $[p]$. For any two subsets $S, T \subseteq [p]$, we say $S \prec_{\mathrm{grlex}} T$ if either*

1. *$|S| < |T|$, or*

2. *$|S| = |T|$ and, when listing the elements of $S$ and $T$ in ascending order under $\pi$, the first index at which they differ belongs to $S$.*

Under this rule, all subsets are grouped by increasing cardinality, and ties are broken by the usual lex order induced by $\pi$.

**Example.** Take $p = 3$ and the identity order $\pi = (1, 2, 3)$. Then the graded–lexicographic sequence of subsets is

$$\emptyset, \{1\}, \{2\}, \{3\}, \{1, 2\}, \{1, 3\}, \{2, 3\}, \{1, 2, 3\}. \quad (79)$$

---

**Algorithm 1:** RECOVERPARENTS($\boldsymbol{p}, \pi, j$)

1: $\mathrm{PA}(\pi(j)) \leftarrow \{\}$, and $X_{\pi,j} \leftarrow (X_{\pi(1)}, \dots, X_{\pi(j)})$
2: Compute the general parameters $\boldsymbol{p}_{\pi,j}$ of $X_{\pi,j}$ // Compute the marginal distribution of $X_{\pi,j}$
3: Convert $\boldsymbol{p}_{\pi,j}$ to natural parameters $\boldsymbol{f}_{\pi,j}$ utilizing Lemma 4
4: **for** $i = 1, \dots, j-1$ **do**

     **if** $\sum_{S \subseteq [\pi(1), \dots, \pi(j-1), \pi(j)] \setminus [\pi(i), \pi(j)]]} \left( f_{\pi,j}^{\pi(i),\pi(j),S} \right)^2 > 0$ **then**

        $\mathrm{PA}(\pi(j)) \leftarrow \mathrm{PA}(\pi(j)) \cup \{\pi(i)\}$
5: **return** $(\mathrm{PA}(\pi(j)), \boldsymbol{f}_{\pi,j})$

---

**Algorithm 2:** RECOVERDAG($\boldsymbol{p}, \pi$)

**Input:** Probability vector $\boldsymbol{p}$ (or empirical count) and topological sort $\pi$
**Output:** DAG $G_\pi$, and natural parameters $\boldsymbol{f}_\pi$
1   $G_\pi \leftarrow$ empty graph and $\boldsymbol{f}_\pi \leftarrow \{\}$
2   **for** $j = 1, 2, \dots, p$ **do**
3     $(\mathrm{PA}(\pi(j)), \boldsymbol{f}_{\pi,j}) \leftarrow$ RECOVERPARENTS($\boldsymbol{p}, \pi, j$)// $\mathrm{PA}(\pi(j))$: the parents of node $\pi(j)$
4     **for** $i \in \mathrm{PA}(\pi(j))$ **do**
5       Add edge $X_i \rightarrow X_j$ to $G_\pi$
6     $\boldsymbol{f}_\pi \leftarrow \boldsymbol{f}_\pi \cup \{\boldsymbol{f}_{\pi,j}\}$

---

Accordingly, the extended feature map $\Phi(X) = [B^S(X)]_{S \subseteq [3]}$ becomes

$$\Phi(X) = \begin{bmatrix} 1, & X_1, & X_2, & X_3, & X_1X_2, & X_1X_3, & X_2X_3, & X_1X_2X_3 \end{bmatrix}. \tag{80}$$

Similarly, for any $j$ and order $\pi$, the parameter block $\boldsymbol{f}_{\pi,j} \in \mathbb{R}^{2^{j-1}}$ is arranged so that its entries align one-to-one with $\Phi(X_{\pi(1)}, \dots, X_{\pi(j-1)})$ in graded–lexicographic order.

### C.4   Procedure for recovering causal graph and parameters

To formalize the recovery procedure from Section 4.2, we present Algorithms 1 and 2.

**Algorithm 1: Parent and parameter recovery.** For a fixed topological order $\pi$ and node index $j \in [p]$, this algorithm

1. computes the marginal probabilities $\boldsymbol{p}_{\pi,j}$ of $(X_{\pi(1)}, \dots, X_{\pi(j)})$,

2. converts $\boldsymbol{p}_{\pi,j}$ to the natural–parameter block $\boldsymbol{f}_{\pi,j}$ via Lemma 4, and

3. selects the parent set $\mathrm{PA}_\pi(j)$ using the nonzero–coefficient criterion in (4).

For estimating the natural-parameter block $\boldsymbol{f}_{\pi,j}$, Section 4.2 uses a logistic regression approach [24]. In Algorithm 1, we instead compute $\boldsymbol{f}_{\pi,j}$ by applying the mapping of Lemma 4 to the marginal probabilities. Under the positivity assumption $\boldsymbol{p} > 0$, these two methods are equivalent and yield identical estimates for $\boldsymbol{f}_{\pi,j}$.

**Algorithm 2: Equivalence-class enumeration.** This algorithm iterates over all $p!$ permutations $\pi \in \mathfrak{G}_p$ and, for each, calls Algorithm 1 for every $j = 1, \dots, p$. It assembles the corresponding DAG $G_\pi$ and parameter collection $\boldsymbol{f}_\pi = \{\boldsymbol{f}_{\pi,j}\}_{j=1}^p$. The output is the full equivalence class

$$\mathcal{E}(\boldsymbol{p}) \;=\; \big\{ (\boldsymbol{f}_\pi, G_\pi) : (\boldsymbol{f}_\pi, G_\pi) \text{ is returned for some } \pi \big\}.$$

Although we state the algorithm in terms of the population vector $\boldsymbol{p}$, in practice one can simply input the data matrix $\mathbf{X} \in \mathbb{R}^{n \times p}$, since empirical frequencies convert $\mathbf{X}$ into $\boldsymbol{p}$.

Building on Algorithms 1 and 2, Algorithm 3 then enumerates all graph–parameter pairs in $\mathcal{E}(\boldsymbol{p})$ and retains only those with the fewest edges, thereby recovering the minimal equivalence class $\mathcal{E}_{\min}(\boldsymbol{p})$ as defined in (8).

**Algorithm 3:** RECOVERSPARSESTDAG($\boldsymbol{p}$)

**Input:** Probability vector $\boldsymbol{p}$ (empirical count)

1 $\mathcal{S} \leftarrow \{\}$
2 **for** *each* $\pi \in \mathfrak{G}_p$ **do**
3 $\quad$ $(G_\pi, \boldsymbol{f}_\pi) \leftarrow$ RECOVERGRAPH$(\boldsymbol{p}, \pi)$// $\mathfrak{G}_p$: set of all the permutation on $p$ variables
4 $\quad$ $\mathcal{S} \leftarrow \mathcal{S} \cup \{(G_\pi, \boldsymbol{f}_\pi)\}$
5 **return** $((G_\pi, \boldsymbol{f}_\pi) \in \mathcal{S} : s_{G_\pi} \leq s_{G_{\tilde{\pi}}}, \forall (G_{\tilde{\pi}}, \boldsymbol{f}_{\tilde{\pi}}) \in \mathcal{S})$

## C.5 Structural equation model

An structural equation model (SEM) [35] $(X, f, P(N))$ over the random vector $X = (X_1, \ldots, X_p)$ is a collection of $p$ structural equations of the form:

$$X_j = f_j(X, N_j), \quad \partial_k f_j = 0 \text{ if } k \notin \text{PA}(j), \tag{81}$$

where $f = (f_j)_{j=1}^p$ is a collection of functions $f_j : \mathbb{R}^{p+1} \to \mathbb{R}$, here $N = (N_1, \ldots, N_p)$ is a vector of independent noises with distribution $P(N)$, and $\text{PA}(j)$ denotes the set of parents of node $j$. Here, $\partial_k f_j$ denotes the partial derivative of $f_j$ w.r.t. $X_k$, which is identically zero when $f_j$ is independent of $X_k$, i.e. $f_j(X, N_j) = f_j(X_{\text{PA}(j)}, N_j)$. The graphical structure induced by the SEM, assumed to be a DAG, will be represented by the following $p \times p$ weighted adjacency matrix $B$:

$$B = B(f), \qquad B_{ij} = \|\partial_i f_j\|_2, \tag{82}$$

and we use $G(B)$ to denote the corresponding binary adjacency matrix.

The structural–equation framework in (81) provides a fully generative foundation for causal discovery by specifying, for each variable, an explicit functional dependence on its direct causes. Its generality encompasses a wide range of models—linear SEMs, additive-noise models, post-nonlinear models, generalized linear models, and more expressive non-linear SEMs. A major virtue of SEMs is their universality: *any* joint distribution can be represented in the form (81) ([35, Prop. 7.1]). In practice, however, researchers impose strong parametric restrictions—such as linearity, additivity, or low-order interactions—that may fail to capture the full complexity of real-world data.

## C.6 Faithfulness, Sparsest Markov representation, Markov equivalence class

We formally define the concepts mentioned in Section 5.

**Definition 4** (Faithfulness [35])**.** *A pair* $(G, P)$ *is said to be* faithful *if:*

$$X_i \perp\!\!\!\perp X_j \mid X_K \quad \Longleftrightarrow \quad X_i \text{ and } X_j \text{ are } d\text{-separated by } X_K \text{ in } G, \tag{83}$$

*for all disjoint subsets* $\{i, j\}, K \subseteq V$. *That is, every conditional independence in* $P$ *corresponds exactly to a* $d$-*separation in* $G$, *and vice versa.*

**Definition 5** (Markov Equivalence Class [42])**.** *Two DAGs* $G_1$ *and* $G_2$ *on the same vertex set* $V$ *are* Markov equivalent *if they encode the same set of conditional independence relations—equivalently, they have the same skeleton (undirected edges) and the same set of v-structures (induced subgraphs of the form* $i \to k \leftarrow j$ *with* $i$ *and* $j$ *not adjacent). The* Markov equivalence class *of a DAG* $G$ *is*

$$\mathcal{M}(G) = \{G' : G' \text{ is a DAG and } G' \text{ is Markov equivalent to } G\}. \tag{84}$$

**Definition 6** (Sparsest Markov Representation [38])**.** *A pair* $(G^0, P)$ *satisfies the* Sparsest Markov Representation (SMR) *assumption if:*

1. *$(G^0, P)$ satisfies the* Markov property*, i.e. every $d$-separation in $G^0$ implies the corresponding conditional independence in $P$.*

2. *For any other DAG $G \notin \mathcal{M}(G^0)$ satisfying the Markov property with respect to $P$, we have*

$$|E(G)| > |E(G^0)|.$$

*Equivalently, $G^0$ is the (unique up to Markov equivalence) sparsest DAG compatible with $P$.*

## C.7 Derivation of the Logistic Loss

First, consider a single example $(X, y)$ with feature vector $X \in \mathbb{R}^m$, binary label $y \in \{0, 1\}$, and parameter vector $w \in \mathbb{R}^m$. Let

$$q = \sigma(w^\top X) = \frac{1}{1 + \exp(-w^\top X)}. \tag{85}$$

The (negative) log-likelihood is

$$\begin{aligned}
\ell(w; X, y) &= -\log\big(q^y (1-q)^{1-y}\big) \\
&= -y \log q - (1-y) \log(1-q).
\end{aligned} \tag{86}$$

Noting that

$$\log \frac{q}{1-q} = w^\top X, \qquad \log(1-q) = -\log\big(1 + \exp(w^\top X)\big), \tag{87}$$

we obtain the familiar logistic-loss form:

$$\ell(w; X, y) = \log\big(1 + \exp(w^\top X)\big) - y\,(w^\top X). \tag{88}$$

In our setting, each "feature" is replaced by the *extended* feature matrix $\Phi(\mathbf{X}) \in \mathbb{R}^{n \times 2^p}$, and each "label" is one column of the data matrix $\mathbf{X} \in \mathbb{R}^{n \times p}$. Stacking over all $p$ columns and averaging over $n$ samples yields

$$\begin{aligned}
\ell(\mathbf{H}; \mathbf{X}) &= \frac{1}{n} \sum_{j=1}^{p} \sum_{i=1}^{n} \Big[ \log\big(1 + \exp[\Phi(\mathbf{X})\mathbf{H}]_{ij}\big) - \mathbf{X}_{ij} \big[\Phi(\mathbf{X})\mathbf{H}\big]_{ij} \Big] \\
&= \frac{1}{n} \sum_{j=1}^{p} \mathbf{1}_n^\top \Big( \log\big(\mathbf{1}_n + \exp(\Phi(\mathbf{X})\mathbf{H})\big) - \mathbf{X}_j \circ (\Phi(\mathbf{X})\mathbf{H}) \Big),
\end{aligned} \tag{89}$$

where $\circ$ denotes the Hadamard product and $\mathbf{1}_n \in \mathbb{R}^n$ is the all-ones vector.

## C.8 Theoretical justification for previous works

Under Assumption A, there exists a topological ordering $\pi$ consistent with $G$ such that each

$$X_{\pi(j)} \text{ is generated by a linear combination of } (X_{\pi(1)}, \ldots, X_{\pi(j-1)}) \in \mathbb{R}^{j-1} \text{ via the logistic link,} \tag{90}$$

rather than via the logistic link on $\Phi\big((X_{\pi(1)}, \ldots, X_{\pi(j-1)})\big) \in \mathbb{R}^{2^{j-1}}$. Importantly, Algorithms 1 and 2 remain valid under this assumption. For any other topological sort $\tilde{\pi} \neq \pi$, the output $(G_{\tilde{\pi}}, f_{\tilde{\pi}})$ from Algorithm 2 may include higher-order interaction terms; nevertheless, by the structural equation model (7), it still recovers the exact distribution

$$X \sim \mathrm{MultiBernoulli}(\boldsymbol{p}), \tag{91}$$

so Theorem 1 continues to hold. Moreover, under Assumption A we can reduce the dimensionality of the optimization (13) from $\mathbf{H} \in \mathbb{R}^{2^p \times p}$ to $\mathbf{H} \in \mathbb{R}^{(p+1) \times p}$. We formalize this reduction below.

Define the parameter matrix

$$\mathbf{H}_j = (\underbrace{h^{0,j}}_{\text{constant}}, \underbrace{h^{1,j}, \ldots, h^{p,j}}_{\text{first order}})^\top \in \mathbb{R}^{p+1} \qquad \mathbf{H} = (\mathbf{H}_1, \ldots, \mathbf{H}_p) \in \mathbb{R}^{(p+1) \times p} \tag{92}$$

where $h^{0,j}$ is the intercept and $h^{1,j}, \ldots, h^{p,j}$ are the first-order coefficients.

The induced adjacency matrix is

$$[W(\mathbf{H})]_{ij} = |h^{i,j}| \tag{93}$$

Self-loops are forbidden, so we impose

$$h^{j,j} = 0 \qquad \forall j \in [p] \tag{94}$$

Redefine the feature map row-wise as

$$\Phi(X) = \begin{bmatrix} 1, & X_1, & \ldots, & X_p \end{bmatrix}, \tag{95}$$

so that for a data matrix $\mathbf{X} \in \mathbb{R}^{n \times p}$, $\Phi(\mathbf{X})$ applies $\Phi$ to each row.

The score (negative log-likelihood) remains

$$\ell(\mathbf{H}; \mathbf{X}) = \frac{1}{n} \sum_{i=1}^{p} \mathbf{1}_n^\top \left( \log(\mathbf{1}_n + \exp(\Phi(\mathbf{X})\mathbf{H})) - \mathbf{X}_i \circ (\Phi(\mathbf{X})\mathbf{H}) \right) \tag{96}$$

We continue to use the quasi-MCP penalty [15], defined by

$$\text{quasi-MCP:} \quad p_{\lambda,\delta}(t) = \lambda \left[ \left( |t| - \frac{t^2}{2\delta} \right) \mathbb{1}\left( |t| < \delta \right) + \frac{\delta}{2} \mathbb{1}\left( |t| > \delta \right) \right] \tag{97}$$

Our final score function is as below

$$s(\mathbf{H}; \lambda, \delta, \mathbf{X}) = s(\mathbf{H}; \mathbf{X}) + p_{\lambda,\delta}(W(\mathbf{H})) \tag{98}$$

We formulate this task as the single continuous optimization problem

$$\min_{\mathbf{H}} \quad s(\mathbf{H}; \lambda, \delta, \mathbf{X})$$
$$\text{subject to} \quad h(W(\mathbf{H})) = 0 \tag{99}$$
$$h^{j,j} = 0 \quad \forall j \in [p]$$

Define the global optimal solution of (99) as

$$\mathcal{O}_{n,\lambda,\delta}^{linear} = \{(\mathbf{H}^*, G(W(\mathbf{H}^*))) : \mathbf{H}^* \text{ is a minimizer of (99)}\} \tag{100}$$

Let $\mathcal{O}_{\infty,\lambda,\delta}^{linear}$ denote the set of minimizers of (99) when the empirical loss $s(\mathbf{H}; \lambda, \delta, \mathbf{X})$ is replaced by its population counterpart $\mathbb{E}\left[ s(\mathbf{H}; \lambda, \delta, \mathbf{X}) \right]$. Let us collect all the parameters in assumption A.

$$H^0 = \begin{bmatrix} c_1 & \cdots & c_p \\ w_1 & \cdots & w_p \end{bmatrix} \in \mathbb{R}^{p+1 \times p} \tag{101}$$

**Theorem 4.** *Suppose Assumption A holds, then $X \sim \text{MultiBernoulli}(\boldsymbol{p})$ where $\boldsymbol{p} > 0$. Moreover, there exist $\lambda, \delta > 0$ sufficiently small such that $(H^0, G^0) \in \mathcal{O}_{\infty,\lambda,\delta}^{linear}$ where $G^0$ is the ground truth graph in Assumption A.*

It is important to note that under this assumption

$$\mathcal{O}_{\infty,\lambda,\delta}^{\text{linear}} \neq \mathcal{E}_{\min}(\boldsymbol{p}), \tag{102}$$

because there may exist a topological sort $\pi$ for which $(\boldsymbol{f}_\pi, G_\pi) \in \mathcal{E}_{\min}(\boldsymbol{p})$ involves higher-order terms, whereas every solution in $\mathcal{O}_{\infty,\lambda,\delta}^{\text{linear}}$ contains only first-order terms.

# D   Experiments

In this section, we present comprehensive experimental details, including the graph types evaluated, the data-generation process, the baseline methods for comparison, the steps required to reproduce our implementation, and the evaluation metrics employed.

## D.1   Experimental Setting

In this section, we outline the process for generating graphs and data. For each model, a random graph $G$ is generated using one of two types of random graph models: Erdős-Rényi(ER) or Scale-Free (SF). The models are specified to have, on average, $kp$ edges, where $k \in \{1, 2, 4\}$. These configurations are denoted as ER$k$ or SF$k$, respectively.

- *Erdős-Rényi*(ER), Random graphs whose edges are add independently with equal probability. We simulated models with $p, 2p$ and $4p$ edges (in expectation) each, denoted by $ER1, ER2$, and $ER4$ respectively.
- Scale-free network(SF). Network simulated according to the preferential attachment process. We simulated scale-free network with $p, 2p$ and $4p$ edges and $\beta = 1$, where $\beta$ is the exponent used in the preferential attachment process.

---

**Algorithm 4:** Generate data matrix $\mathbf{X}$

---

**Input:** DAG $G$, sample size $n$, interaction type $\tau \in \{$1st+2nd, 1st+pth, 1st+2nd+pth 2nd, pth$\}$
**Output:** $\mathbf{X} \in \{0,1\}^{n \times p}$

1   Compute a topological ordering $\pi$ of $G$
2   **for** $j \leftarrow 1$ **to** $p$ **do**
3      Sample $w_{\mathrm{PA}(\pi(j))} = (w_k)_{k=1}^{2^{|\mathrm{PA}(\pi(j))|}} \in \mathbb{R}^{2^{|\mathrm{PA}(\pi(j))|}}, \quad w_k \overset{\mathrm{iid}}{\sim} \mathrm{Unif}([-2,-1] \cup [1,2])$.
4      **if** $\tau = $ *1st+2nd* **then**
5         $q \leftarrow \sigma\big(w_{\mathrm{PA}(\pi(j))}^\top \Phi^{\mathrm{1st+2nd}}(\mathbf{X}_{\mathrm{PA}(\pi(j))})\big)$
6      **else if** $\tau = $ *1st+pth* **then**
7         $q \leftarrow \sigma\big(w_{\mathrm{PA}(\pi(j))}^\top \Phi^{\mathrm{1st+pth}}(\mathbf{X}_{\mathrm{PA}(\pi(j))})\big)$
8      **else if** $\tau = $ *2nd* **then**
9         $q \leftarrow \sigma\big(w_{\mathrm{PA}(\pi(j))}^\top \Phi^{\mathrm{2nd}}(\mathbf{X}_{\mathrm{PA}(\pi(j))})\big)$
10     **else**
11        $q \leftarrow \sigma\big(w_{\mathrm{PA}(\pi(j))}^\top \Phi^{\mathrm{pth}}(\mathbf{X}_{\mathrm{PA}(\pi(j))})\big)$
12     $\mathbf{X}_{\pi(j)} \sim \mathrm{Bernoulli}(q)$

---

**General binary data**   Since we wish to study structure learning for general binary data, Theorem 1 implies that for any $\boldsymbol{p} > 0$,

$$X \sim \mathrm{MultiBernoulli}(\boldsymbol{p}) \tag{103}$$

can be generated via the SEM (7). To allow different interaction orders, define the following extended feature maps for $X = (X_1, \ldots, X_p)$:

$$\Phi^{\mathrm{1st+2nd}}(X) = (\underbrace{1}_{\text{constant}}, \underbrace{X_1, \ldots, X_p}_{\text{first order}}, \underbrace{X_1 X_2, \ldots, X_{p-1} X_p}_{\text{second order}}, \underbrace{0}_{\text{third order}}, \underbrace{0}_{\text{forth to } (p-1)\text{-th order}}, \underbrace{0}_{p\text{-th order}})^\top \in \mathbb{R}^{2^p}$$

$$\Phi^{\mathrm{1st+2nd+pth}}(X) = (\underbrace{1}_{\text{constant}}, \underbrace{X_1, \ldots, X_p}_{\text{first order}}, \underbrace{X_1 X_2, \ldots, X_{p-1} X_p}_{\text{second order}}, \underbrace{0}_{\text{third order}}, \underbrace{0}_{\text{fourth to } (p-1)\text{-th order}}, \underbrace{X_1 \ldots X_p}_{p\text{-th order}})^\top \in \mathbb{R}^{2^p}$$

$$\Phi^{\mathrm{1st+pth}}(X) = (\underbrace{1}_{\text{constant}}, \underbrace{X_1, \ldots, X_p}_{\text{first order}}, \underbrace{0}_{\text{second order}}, \underbrace{0}_{\text{third order}}, \underbrace{0}_{\text{forth to } (p-1)\text{-th order}}, \underbrace{X_1 X_2 \ldots X_p}_{p\text{-th order}})^\top \in \mathbb{R}^{2^p}$$

$$\Phi^{\mathrm{2nd}}(X) = (\underbrace{1}_{\text{constant}}, \underbrace{0}_{\text{first order}}, \underbrace{X_1 X_2, \ldots, X_{p-1} X_p}_{\text{second order}}, \underbrace{0}_{\text{third order}}, \underbrace{0}_{\text{forth to } (p-1)\text{-th order}}, \underbrace{0}_{p\text{-th order}})^\top \in \mathbb{R}^{2^p}$$

$$\Phi^{\mathrm{pth}}(X) = (\underbrace{1}_{\text{constant}}, \underbrace{0}_{\text{first order}}, \underbrace{0}_{\text{second order}}, \underbrace{0}_{\text{third order}}, \underbrace{0}_{\text{forth to } (p-1)\text{-th order}}, \underbrace{X_1 X_2 \ldots X_p}_{p\text{-th order}})^\top \in \mathbb{R}^{2^p} \tag{104}$$

where in each vector the nonzero blocks correspond respectively to the constant term, first-order terms, second-order terms, and highest-order term. By convention, if $X = \emptyset$, then all four maps reduce to the scalar $(1) \in \mathbb{R}$. When applied to a data matrix $\mathbf{X} \in \mathbb{R}^{n \times p}$, each $\Phi(\mathbf{X})$ operates row-wise.

Finally, given a random DAG $B \in \{0,1\}^{p \times p}$ sampled from one of our graph models, we generate $\mathbf{X}$ using Algorithm 4, choosing the desired interaction map according to whether we study first+second, first+highest, second, or highest-order interactions.

**Simulation**   We generate random dataset $\mathbf{X} \in \mathbb{R}^{n \times p}$ by sampling i.i.d from the models described above. For each simulation, we produce datasets with $n$ samples cross graphs with $p$ nodes.

- **(Small Graph)** $p = \{5, 6, 7, 8, 9\}$, $k = \{1, 2\}$, $n = 10000$ and graph types: $\{$ER,SF$\}$
- **(Large Graph)** $p = \{10, 20, 30, 40\}$, $k = \{1, 2, 4\}$, $n = 1000$ and graph types: $\{$ER,SF$\}$

## D.2 Implementation

For each dataset, we applied several structural learning algorithms, including fast greedy equivalence search (FGES [37]), constraint-based methods (PC [42]), DAGMA [5], NOTEARS [55]. The implementation details are provided in the following paragraph. After running the algorithms, a post-processing threshold of $0.3$ was applied to the estimated $B_{\text{est}}$ to prune small values, following the same procedure in [54, 5].

- Fast Greedy Equivalence Search (FGES [37]) is based on greedy search and assumes linear dependency between variables. The implementation is based on the `py-tetrad` package, available at https://github.com/cmu-phil/py-tetrad. We use `search.use_bdeu(sample_prior=10, structure_prior=0)`.

- PC [42] is constraint-based method and based on uses conditional independence induced by causal relationships to learn those causal relationships. The implementation is based on the `py-tetrad` package, available at https://github.com/cmu-phil/py-tetrad. We use `search.use_chi_square(alpha=0.1)`

- NOTEARS-MLP [55] is a continuous DAG-learning method that employs a least-squares loss with $\ell_1$ regularization. Its Python implementation is available at https://github.com/xunzheng/notears.

- DAGMA [5] is a continuous DAG-learning algorithm that achieves improved accuracy and faster computation, with barrier methods. Its implementation can be found at https://github.com/kevinsbello/dagma. We use the default parameters from original implementation.

- BiNOTEARS builds on DAGMA and NOTEARS-MLP, with the choice of solver tailored to problem size. For small graphs ($d \in \{5, 6, 7, 8, 9\}$), we optimize the full formulation (13)—including all higher-order interaction terms—using the DAGMA framework. For larger graphs ($d \in \{10, 20, 30, 40\}$), we adopt the NOTEARS-MLP framework. The original implementation optimizes an $\ell_2$ loss with an $\ell_1$ penalty; in our variant, we insert a sigmoid activation $\sigma(x) = 1/(1 + \exp(-x))$ on the final layer and replace the squared loss with the cross-entropy (logistic) loss to accommodate binary data. After estimating the weighted adjacency $B_{\text{est}}$ via NOTEARS-MLP, we prune entries below $0.3$, compute a topological ordering of the resulting graph, and then apply Algorithm 2 with first- and second-order terms to obtain the final structure. Finally, we remove any remaining edges whose weight does not exceed $1.0$ to eliminate spurious connections.

**Hyperparameter tuning**  Theorem 3 indicates that one should ideally choose small values of $\lambda$ and $\delta$ for the quasi-MCP penalty. In practice, however, achieving the global optimum of (13) is infeasible, and if $\lambda$ and $\delta$ are too small the penalty becomes ineffective and the algorithm may fail to recover the sparsest solution. To mitigate this, we adopt the continuation strategy of Deng et al. [15]: start with relatively large $\lambda$ and $\delta$, solve (13) via BiNOTEARS to obtain an initial estimate $\mathbf{H}_{\text{est}}$, then iteratively shrink $\lambda$ and $\delta$ by a factor $\gamma < 1$, using the previous estimate as the warm start for the next run of BiNOTEARS. We terminate when the negative log-likelihood $s(\mathbf{H}_{\text{est}}; \mathbf{X})$ ceases to decrease. Empirically, $\gamma = 0.5$, $\lambda = 0.05$, and $\delta = 0.2$ perform well in our experiments.

**Equipment**  The experiments are conducted in the following CPU architectures

- Intel Broadwell—28 cores @ 2.4 GHz with 64 GB memory per node

- Intel Skylake—40 cores @ 2.4 GHz with 96 GB memory per node

## D.3 Metrics

- **Structural Hamming distance (SHD)**: A standard benchmark in the structure learning literature that counts the total number of edges additions, deletions, and reversals needed to convert the estimated graph into the true graph. Since our data specified in (1) is nonidentifiable, the Structural Hamming Distance (SHD) is calculated with respect to the completed partially directed acyclic graph (CPDAG) of the ground truth and $B_{\text{est}}$.

# E  Additional Figures

## E.1  Small graphs

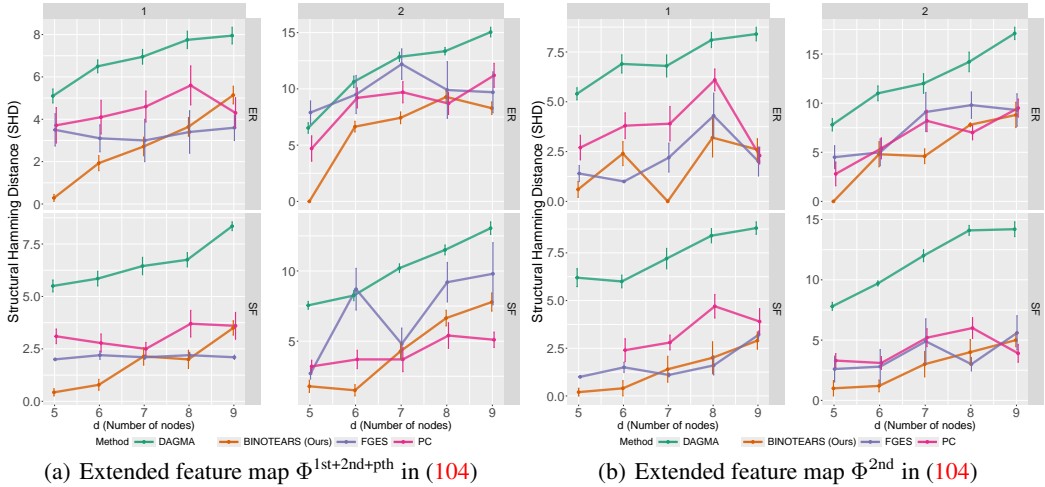

(a) Extended feature map $\Phi^{\text{1st+2nd+pth}}$ in (104)    (b) Extended feature map $\Phi^{\text{2nd}}$ in (104)

Figure 3: Results in terms of SHD between MECs of estimated graph and ground truth. Lower is better. Column: $k = \{1, 2\}$. Row: random graph types. $\{\text{ER,SF}\}$-$k = \{\text{Scale-Free}, \text{Erdős–Rényi}\}$ graphs with $kd$ expected edges. Here $p = \{5, 6, 7, 8, 9\}$. Error bars denote the standard error computed over 10 replications.

## E.2  Large graphs

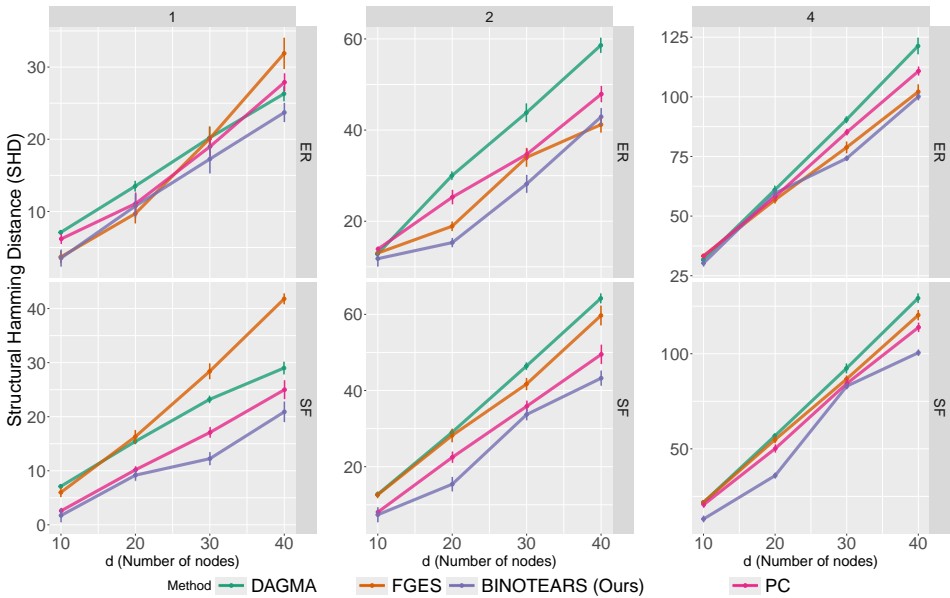

Figure 4: Results in terms of SHD between MECs of estimated graph and ground truth. Lower is better. Data are generated using extended feature map $\Phi^{\text{1st+pth}}$ in (104). Column: $k = \{1, 2, 4\}$. Row: random graph types. $\{\text{ER,SF}\}$-$k = \{\text{Scale-Free}, \text{Erdős–Rényi}\}$ graphs with $kd$ expected edges. Here $p = \{10, 20, 30, 40\}$. BıNOTEARS is our two stage approach.

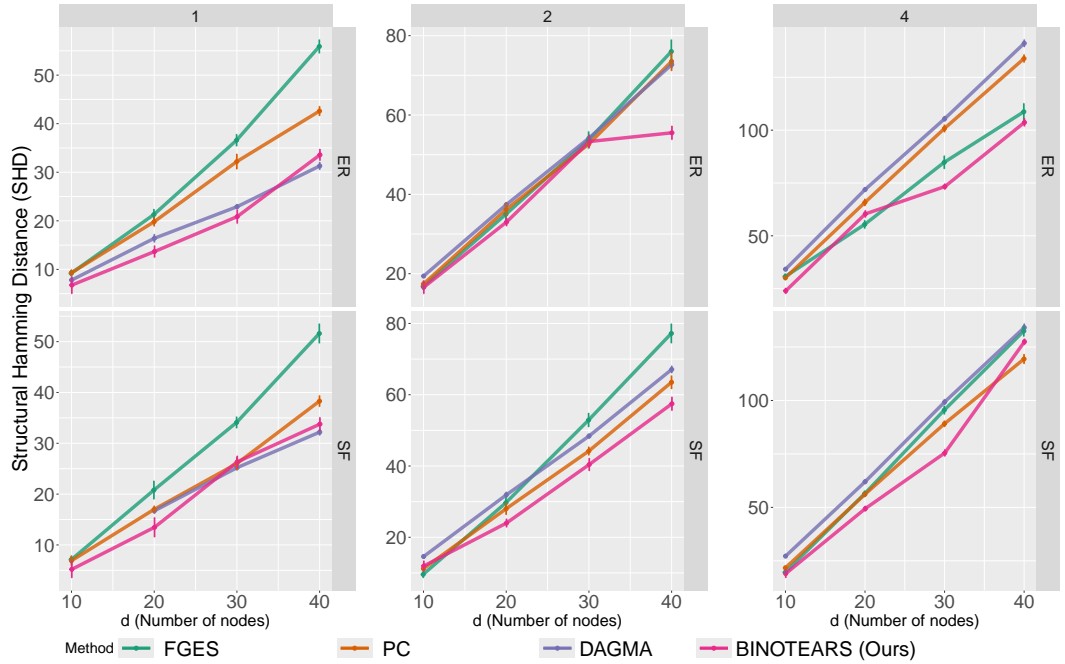

Figure 5: Results in terms of SHD between MECs of estimated graph and ground truth. Lower is better. Data are generated using extended feature map $\Phi^{\text{2nd}}$ in (104). Column: $k = \{1, 2, 4\}$. Row: random graph types. $\{\text{ER,SF}\}\text{-}k = \{\text{Scale-Free}, \text{Erdős–Rényi}\}$ graphs with $kd$ expected edges. Here $p = \{10, 20, 30, 40\}$. BINOTEARS is our two stage approach.

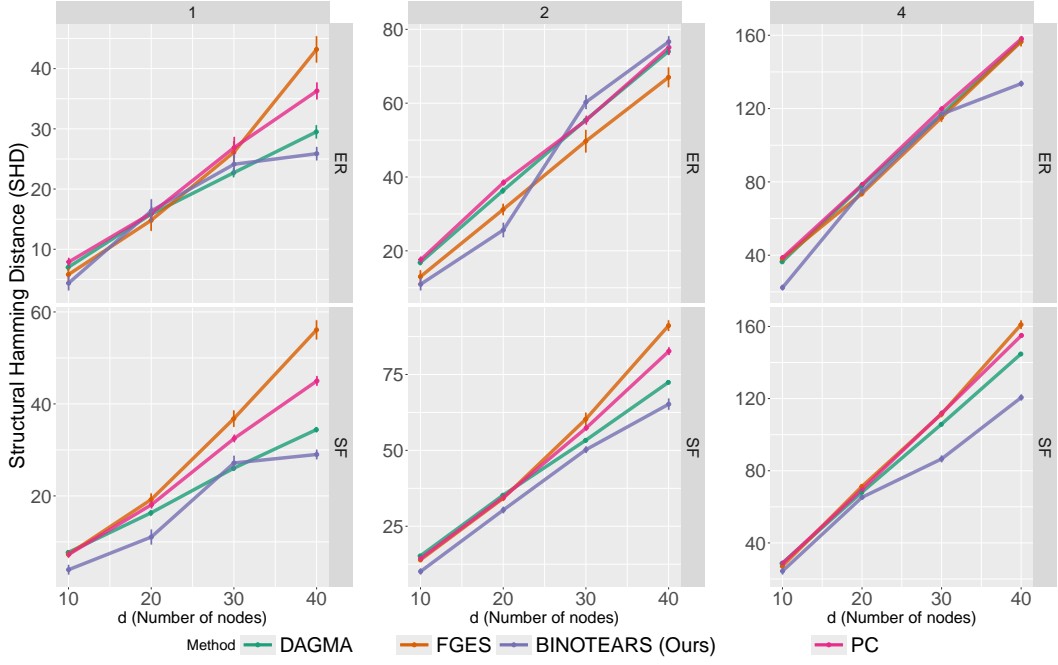

Figure 6: Results in terms of SHD between MECs of estimated graph and ground truth. Lower is better. Data are generated using extended feature map $\Phi^{\text{pth}}$ in (104). Column: $k = \{1, 2, 4\}$. Row: random graph types. $\{\text{ER,SF}\}\text{-}k = \{\text{Scale-Free}, \text{Erdős–Rényi}\}$ graphs with $kd$ expected edges. Here $p = \{10, 20, 30, 40\}$. BINOTEARS is our two stage approach.

