# OpenReview forum: "Differentiable Structure Learning and Causal Discovery for General Binary Data"
_NeurIPS.cc/2025/Conference — NeurIPS 2025 poster_

### Official Review · Reviewer_WmEn · 2025-06-24

**Clarity:** 4
**Significance:** 3
**Originality:** 3
**Rating:** 5
**Confidence:** 4

**Summary:**

The paper proposes a general differential structure learning framework and continuous DAG learning algorithm for discrete data that could capture arbitrary dependencies with no distribution restrictions. To address the combinatorial search problems in discrete data, the paper proposes Notears-MLP-Reg method to reduce computational complexity.

**Questions:**

See above.

**Ethical Concerns:**

["NO or VERY MINOR ethics concerns only"]

**Final Justification:**

I believe my concerns are resolved especially around computational time.

**Limitations:**

Yes.

**Paper Formatting Concerns:**

No issues identified.

**Quality:**

3

**Strengths And Weaknesses:**

The paper is very well-written with clear structured theoretical investigation and empirical evidence. The novelty is to spot the equivalence of modelling binary data as exponential families with interaction terms as potential candidates for causal parents.

Comments:
- in the paper there are multiple places referring that general discrete models are unidentifiable from purely observational data alone (e.g. in abstract line 8, line 218, line 222) and there are also places restricting to the general multivariate Bernoulli distribution (see line 62, line 245) that indicates single environement /  distribution rather than observational data alone. However, it is well-known that multi-environment observational data alone could identify causal structure [1, 2]. To make the paper more precise, it would be good to clarify the differences.

- Theorem 3 says there exists $\lambda$, $\delta$ sufficiently small such that the predicted set is the same as minimal equivalence set. Could the authors discuss in practice (when performing experiments) how to select $\lambda$, $\delta$. Is it similar as the theorem suggests, the smaller the better?

- The natural problem in discrete data is combinatorial search. The authors address this problem via proposing NoTEARS-MLP-Reg - which first extracts topological ordering and then fit interaction features consistent with $\pi$ and restrict to first and second order. Could the authors clarify, under this setup, how did the number of searches scale in terms of number of node increases?

[1] S. Guo, V. Toth, B. Schölkopf, and F. Huszar. Causal de Finetti: On the Identification of Invariant Causal Structure in Exchangeable Data. In A. Oh, T. Neumann, A. Globerson, K. Saenko, M. Hardt, and S. Levine, editors, Advances in Neural Information Processing Systems, volume 36, pages 36463–36475. Curran Associates, Inc., 2023a

[2] Huang, B., Zhang, K., Zhang, J., Ramsey, J., Sanchez-Romero, R., Glymour, C., & Schölkopf, B. (2020). Causal discovery from heterogeneous/nonstationary data. Journal of Machine Learning Research, 21(89), 1-53.

---

> ### Author Rebuttal · Authors · 2025-07-30
>
> We thank the reviewer for their time and effort, and for the positive feedback on our paper’s clarity, theoretical guarantees, and empirical evaluation. We will incorporate the following discussion into the revised manuscript.
>
> > in the paper there are multiple places referring that general discrete models are unidentifiable from purely observational data alone (e.g. in abstract line 8, line 218, line 222) and there are also places restricting to the general multivariate Bernoulli distribution (see line 62, line 245) that indicates single environment / distribution rather than observational data alone. However, it is well-known that multi-environment observational data alone could identify causal structure [1, 2]. To make the paper more precise, it would be good to clarify the differences.
> >
>
> Thank you for raising this important point. We will revise the manuscript to make it clear that our identifiability results pertain strictly to the single‑environment, purely observational setting—under which general discrete models remain non‑identifiable. We will then contrast this with the multi‑environment case (e.g. data collected under different interventions), in which causal structure can be identified, as shown in [1,2]. In the updated draft we’ll explicitly distinguish these two scenarios and discuss the relevant literature.
>
> > Theorem 3 says there exists $\lambda,\delta$, sufficiently small such that the predicted set is the same as minimal equivalence set. Could the authors discuss in practice (when performing experiments) how to select $\lambda,\delta$, . Is it similar as the theorem suggests, the smaller the better?
> >
>
> Thank you for this insightful question. The related discussion is included at Appendix D.1 (Lines 1088–1095) in supplementary file due to space limit. In theory, any$(\lambda,\delta)$ below the threshold specified in Theorem 3 will recover the minimal equivalence set. In practice, however, choosing overly small values can trigger numerical‐precision issues, causing the quasi‐MCP penalty to under‑regularize and produce an overly dense graph. To strike the right balance, we employ a simple “weight‑decay” method proposed in [3]:
>
> - **Initialization**: Start with relatively large $\lambda,\delta$ and solve (13) to obtain an initial estimate.
> - **Decay**: Multiply both $\lambda$ and $\delta$ by a factor $\gamma<1$ and re‑solve, using the previous solution as a warm start.
> - **Stopping criterion**: Stop once the negative log‑likelihood ceases to decrease.
>
> Empirically, this procedure yields sparse, stable, and accurate graph estimates.
>
> > The natural problem in discrete data is combinatorial search. The authors address this problem via proposing NoTEARS-MLP-Reg - which first extracts topological ordering and then fit interaction features consistent with and restrict to first and second order. Could the authors clarify, under this setup, how did the number of searches scale in terms of number of node increases?
> >
>
> Thank you for raising this important question. NoTEARS‑MLP‑Reg is a two‑stage procedure. In the first stage, we train an MLP (NOTEARS‑MLP [4]) to capture higher‑order interactions in discrete data and recover an initial graph structure. We then extract a node ordering $\hat{\pi}$ from this preliminary graph and, in the second stage, fit a logistic regression model—including both first‑ and second‑order terms—to recover the final graph. Since logistic regression under a fixed ordering is very efficient, the MLP‑based stage remains the dominant computational cost. Empirically, the overall runtime of NoTEARS‑MLP‑Reg closely matches that of NOTEARS‑MLP [4]. The table below summarizes these runtimes and will be included in the final manuscript.
>
>
> |Times(second)| NOTEARS-MLP-REG |
> |:-----------:|:---------------:|
> | ER2 (d=10)  | 43.223          |
> | ER4 (d=10)  | 38.972          |
> | SF2 (d=10)  | 37.925          |
> | SF4 (d=10)  | 41.259          |
> | ER2 (d=20)  | 84.392          |
> | ER4 (d=20)  | 87.267          |
> | SF2 (d=20)  | 80.952          |
> | SF4 (d=20)  | 93.136          |
> | ER2 (d=40)  |     295.333     |
> | ER4 (d=40)  |     337.205     |
> | SF2 (d=40)  |     293.903     |
> | SF4 (d=40)  |     313.349     |
>
>
> From the reported runtimes, it is clear that our method does not exhibit exponential time complexity and can deliver meaningful discoveries within a reasonable timeframe.
>
> [1] S. Guo, V. Toth, B. Schölkopf, and F. Huszar. Causal de Finetti: On the Identification of Invariant Causal Structure in Exchangeable Data. In A. Oh, T. Neumann, A. Globerson, K. Saenko, M. Hardt, and S. Levine, editors, Advances in Neural Information Processing Systems, volume 36, pages 36463–36475. Curran Associates, Inc., 2023a
>
> [2] Huang, B., Zhang, K., Zhang, J., Ramsey, J., Sanchez-Romero, R., Glymour, C., & Schölkopf, B. (2020). Causal discovery from heterogeneous/nonstationary data. Journal of Machine Learning Research, 21(89), 1-53.
>
> [3] Deng, Chang, et al. "Markov equivalence and consistency in differentiable structure learning." *Advances in Neural Information Processing Systems* 37 (2024)
>
> [4] Zheng, Xun, et al. "Learning sparse nonparametric dags." *International conference on artificial intelligence and statistics*. Pmlr, 2020.

---

> > ### Comment · Reviewer_WmEn · 2025-08-01
> >
> > Thank you for your response and additional experimental results. I will keep my score.

---

> > > ### Author Response · Authors · 2025-08-01
> > >
> > > We thank the reviewer for these valuable suggestions. We will incorporate all points raised in the rebuttal into the revised manuscript. Thank you for your time and effort.

---

### Official Review · Reviewer_1Kmy · 2025-07-01

**Clarity:** 2
**Significance:** 2
**Originality:** 3
**Rating:** 4
**Confidence:** 2

**Summary:**

This paper provides a general differentiable structure learning framework for causal discovery on binary data from the multivariate Bernoulli distribution, embedding all higher-order interactions without restrictive parametric assumptions.
It first provides a non-identifiability result under matching observable distributions.
Then it shows the proposed algorithm identifies the sparest DAG up to Markov equivalence class under faithfulness.
Experiments are conducted on simulated synthetic data, compared with a set of causal discovery baselines.

**Questions:**

1. Following up on the first weakness: it would be helpful to include a real-world motivational example where modeling higher-order interactions is crucial for causal discovery. This would better justify the need for such a general formulation beyond synthetic settings.
2. What is the computational complexity of the proposed DAGMA-HO method, particularly with respect to the number of nodes and the order of interactions?
3. Is it always beneficial to model higher-order interaction terms? How should one choose the order of interaction terms in practice? Is there a principled way to select this, or is it treated as a hyperparameter?

**Ethical Concerns:**

["NO or VERY MINOR ethics concerns only"]

**Final Justification:**

The rebuttal has addressed most of my concerns, and I am happy to increase my score to 4. I kindly encourage the authors to ensure these clarifications are incorporated into the revised manuscript to improve overall clarity.

**Limitations:**

yes

**Paper Formatting Concerns:**

In the main manuscript, every citation of the appendix appears as a “??” placeholder. Although I reviewed the supplementary files to follow the arguments, this formatting error represents a significant flaw in the submission.

**Quality:**

2

**Strengths And Weaknesses:**

### Strengths
1. The paper tackles a broad nonparametric framework that better reflects real-world scenarios, and backs its claims with comprehensive theoretical proofs.
2. The narrative is overall well-structured.
3. On synthetic Erdős–Rényi and scale-free graphs, the method achieves competitive performance compared to existing baselines.

### Weaknesses
1. While modelling higher-order interactions is motivated by real-world use cases, a motivational example is missing. Since the experiments are also focusing on synthetic data, it is not clear to me why higher-order interactions should play an important role in real-world causal discovery tasks.
2. As noted by the authors, the full higher-order feature map scales exponentially with the number of nodes, limiting its applicability to larger networks. The NOTEARS-MLP-REG heuristic helps in practice but lacks theoretical guarantees, making its connection to the paper’s core identifiability results less clear.
3. The notation in Section 4 is difficult to follow. Simplifying or redesigning symbols would greatly improve readability, e.g., by reducing super- and subscripts.
4. Visualization: Please keep the color-coding aligned for all figures in the paper! For example, FEGS was marked as purple in figure 1 but orange in figure 2.

---

> ### Author Rebuttal · Authors · 2025-07-30
>
> We thank the reviewer for their constructive comments and are pleased that they find our paper well‑structured and theoretically comprehensive. We will address all concerns and incorporate their suggestions in the revised manuscript.
>
> > While modelling higher-order interactions is motivated by real-world use cases, a motivational example is missing. Since the experiments are also focusing on synthetic data, it is not clear to me why higher-order interactions should play an important role in real-world causal discovery tasks.
> >
>
> > Following up on the first weakness: it would be helpful to include a real-world motivational example where modeling higher-order interactions is crucial for causal discovery. This would better justify the need for such a general formulation beyond synthetic settings.
> >
>
> Thank you for emphasizing the need for a real‑data evaluation. We applied DAGMA‑HO to the [1] benchmark dataset, which comprises $n=7466$ continuous expression measurements of $d=11$ proteins and phospholipids in human immune cells and includes a consensus network widely accepted by the biological community. We worked with discretized dataset, and ran DAGMA‑HO. Our method recovers the consensus structure with competitive accuracy compared to existing approaches, demonstrating its potential existence of higher order interaction in real data application.
>
>
> |                | SHD | Num of Edges |
> |----------------|:---:|:------------:|
> | NOTEARS-Linear[2] |  22 |      18      |
> | NOTEARS-MLP[3]    |  16 |      13      |
> | DAGMA-HO(Ours) |  13 |      15      |
>
> > As noted by the authors, the full higher-order feature map scales exponentially with the number of nodes, limiting its applicability to larger networks. The NOTEARS-MLP-REG heuristic helps in practice but lacks theoretical guarantees, making its connection to the paper’s core identifiability results less clear.
> >
>
> Thank you for raising this important point. Our manuscript is primarily concerned with the **identifiability** of discrete‐valued DAGs—i.e., with characterizing when the global minimizers of (13) recover the true minimal‐equivalence class under a general binary model. We deliberately *abstract away* the algorithmic question of how to solve (13) at scale, since:
>
> 1. **DAG learning is NP‑hard** in general, and most existing methods resort to heuristics without strong guarantees.
> 2. Our focus is on the properties of the solution set, not on optimization algorithms.
>
> That said, we agree that addressing computational tractability is a crucial—and distinct—research direction. **NoTEARS‑MLP‑REG** represents our first attempt to bridge this gap in practice: by utilizing the neural network to recover graph and extracting a node ordering and then fitting penalized regressions up to second order, we avoid an exponential search over DAG space. Although it does not yet come with a formal consistency proof, it demonstrates that moderate‐size problems are solvable in reasonable time.
>
> We’ll add a paragraph in the discussion to:
>
> 1. Emphasize that identifiability and efficient optimization are orthogonal challenges.
> 2. Position NoTEARS‑MLP‑REG as a *heuristic prototype*, analogous to other NP‑hard DAG‑learning schemes (e.g.,GES, NOTEARS, DAGMA).
> 3. Highlight the open question of designing **provably** efficient algorithms that trade off between (i) restricted model space (for identifiability) such as Assumption A (only first order interaction is allowed) and (ii) polynomial‑time solvability.
>
> > The notation in Section 4 is difficult to follow. Simplifying or redesigning symbols would greatly improve readability, e.g., by reducing super- and subscripts.
> >
>
> We apologize for the heavy notation in Section 4. In the revised manuscript, we will streamline our symbols by reducing unnecessary superscripts and subscripts, especially in Eq (6) where the natural parameter is defined. Concretely, we introduce
>
> $$
> \mathcal{T}_{\pi,j} = \\{T\subseteq\\{\pi(1),\ldots,\pi(j-1)\\} \\}
> $$
>
> and then write the parameter vector as
>
> $$
> \boldsymbol{f}\_{\pi,j} =\left(f\_{T\cup \{\pi(j)\} }\right)\_{T\in\mathcal{T}_{\pi,j}}\in \mathbb{R}^{2^{j-1}}
> $$
>
> or equivalently
>
> $$
> \boldsymbol{f}\_{\pi,j}  = \left(f\_{S}\right)\_{S\subseteq \\{\pi(1),\ldots,\pi(j)\\},\pi(j)\in S }
> $$
>
> Likewise, we define the feature vector
>
> $$
> \mathbf{\Phi}\_{\pi,j}(X) = \left( B_S(X)\right)\_{S\in \mathcal{T}\_{\pi,j}\cup \\{\pi(j)\\} }
> $$
>
> Under this notation, the conditional probability becomes
>
> $$
> P(X\_{\pi(j)}=1\mid X_{\pi(1:j-1)})  =\mathrm{logistic}(\boldsymbol{f}\_{\pi,j}^\top\mathbf{\Phi}\_{\pi,j}(X))
> $$
>
> A similar simplification applies to each $\mathbf{H}_j$. We will also include a simple illustrative example to ensure full rigor while greatly improving clarity.
>
> > Visualization: Please keep the color-coding aligned for all figures in the paper! For example, FEGS was marked as purple in figure 1 but orange in figure 2.
> >
>
> We appreciate this suggestion and will ensure that all color mappings remain consistent.
>
>
>
> > What is the computational complexity of the proposed DAGMA-HO method, particularly with respect to the number of nodes and the order of interactions?
> >
>
> This question is quite interesting. Suppose we have $n$ samples and $p$  variables, and include all interactions up to order $k$, so that the total feature dimension is $\sum_{i=0}^k \binom{p}{i}\approx p^k$. Computing the gradient of the regularized score function then costs $O(n\cdot p\cdot p^k)$ for the score‑function terms plus $O(p^2\cdot p^k)$ for the quasi‑MCP penalty, which together give $O\bigl((n+p)\cdot p^{k+1}\bigr)$. The gradient of the acyclicity constraint $h\bigl(W(H)\bigr)$ requires $O\bigl(p^3 + p^2\cdot p^k\bigr)=O\bigl(p^{k+2}\bigr)$, so a single gradient evaluation remains $O\bigl((n+p)\cdot p^{k+1}\bigr)$. If the DAGMA uses $T_{\rm out}$ outer updates (to decrease the penalty weight) and $T_{\rm inner}$ inner iterations (e.g. Adam steps), the total complexity becomes $O\bigl(T_{\rm inner}\cdot T_{\rm out}\cdot (n+p)\cdot p^{k+1}\bigr)$. In the extreme case where all higher‑order interactions are included (i.e. $k=p$), replacing $p^k$ with $2^p$ yields a total cost of $O\bigl(T_{\rm inner}T_{\rm out}(n+p)p 2^p \bigr).$
>
> > Is it always beneficial to model higher-order interaction terms? How should one choose the order of interaction terms in practice? Is there a principled way to select this, or is it treated as a hyperparameter?
> >
>
> In theory, allowing every possible interaction can only increase the expressiveness of the model—and recovering the true DAG would benefit from including arbitrarily high orders. In practice, however, the number of interaction terms grows combinatorially in the number of nodes $p$, making full higher‐order expansions infeasible except for very small $p$.
>
> Empirically, we and others have found that truncating at second order strikes a reliable balance: it captures most of the “nonlinear” dependencies you’d see in real data, yet keeps the total feature count—and hence both memory and runtime—within practical limits.
>
> When $p$ is very small (say under 10), exploring up to third or even fourth order can be done directly; once $p$ grows beyond a dozen or so, second order is almost always the sweet spot.
>
> [1] Sachs, Karen, et al. ”Causal protein-signaling networks derived from multiparameter single-cell data.” Science 2005.
>
> [2] Zheng, Xun, et al. "Dags with no tears: Continuous optimization for structure learning." *Advances in neural information processing systems* 31 (2018).
>
> [3] Zheng, Xun, et al. "Learning sparse nonparametric dags." *International conference on artificial intelligence and statistics*. Pmlr, 2020.

---

> > ### Comment · Reviewer_1Kmy · 2025-08-04
> >
> > Thank you for the explanation. I appreciate the inclusion of the real-world experiment on [1]—this certainly strengthens the paper. Regarding the motivational example, I was hoping for something more intuition-driven rather than result-oriented. While the empirical findings support the benefit of incorporating higher-order interactions, it remains somewhat unclear—especially from the introduction—why this design choice is intuitively justified. It would be helpful if the authors could include a textual explanation or heuristic motivation to clarify this aspect for readers.
> >
> > That said, the rebuttal has addressed most of my concerns, and I am happy to increase my score to 4. I kindly encourage the authors to ensure these clarifications are incorporated into the revised manuscript to improve overall clarity.

---

> > > ### Author Response · Authors · 2025-08-04
> > > **A real motivation example from epidemiologic literature**
> > >
> > > Thank you for the helpful suggestion and raising our score!!
> > >
> > > We agree that the manuscript would benefit from a more intuition-driven motivation for incorporating higher-order interactions, and we will add a brief, clear explanation in the revised version. In particular, when the topological order is fixed, recovering the DAG reduces to fitting a sequence of logistic regressions (Theorem 1), and in that setting higher-order interaction terms naturally arise as _effect modification_[4]: the influence of one parent on the log-odds can depend on the value of another. This phenomenon is well understood in the epidemiologic literature [4], where interaction terms in logistic models are used to capture departures from simple additivity or multiplicativity—for example, the supra-multiplicative (synergistic) joint effect of smoking and asbestos exposure on lung cancer risk is better represented by including their product term[5].
> > >
> > > Therefore, the presence of such conditional dependence among discrete variables provides a heuristic justification for including higher-order interactions: omitting them would correspond to a misspecified model that fails to capture synergistic or effect-modifying influences. We will incorporate this concise motivation into the introduction to improve clarity.
> > >
> > > Thanks again for your time!
> > >
> > > [4] VanderWeele, Tyler J., and Mirjam J. Knol. "A tutorial on interaction." Epidemiologic methods 3.1 (2014): 33-72.
> > >
> > > [5] Ngamwong, Yuwadee, et al. “Additive Synergism between Asbestos and Smoking in Lung Cancer Risk: A Systematic Review and Meta-Analysis.” PLoS ONE 10, no. 8 (2015): e0135798.

---

### Official Review · Reviewer_QLcb · 2025-07-02

**Clarity:** 3
**Significance:** 2
**Originality:** 2
**Rating:** 4
**Confidence:** 3

**Summary:**

This paper addresses differentiable structure learning for discrete data. Arguing that prior work relies on overly restrictive parametric assumptions, the authors propose a general framework starting from the multivariate Bernoulli (MVB) distribution to capture arbitrary dependencies. This paper proposes finding the sparsest graph, relying on the Sparsest Markov Representation (SMR) assumption to guarantee uniqueness up to the Markov Equivalence Class, and it translates this search into a differentiable optimization problem and introduces a two-stage heuristic to address the primary method's exponential complexity.

**Questions:**

I'd appreciate the authors' feedback on the weakness section.

**Ethical Concerns:**

["NO or VERY MINOR ethics concerns only"]

**Final Justification:**

After reading the authors' response and the other reviewers' comments, I've decided to maintain my current rating.

**Limitations:**

Please see the weakness section.

**Quality:**

3

**Strengths And Weaknesses:**

Strengths:

1. The paper is exceptionally well-written. The argument flows logically from the limitations of prior work to the general theoretical characterization, the proposed solution, and the empirical validation. The core ideas, definitions, and theorems are presented with precision and clarity.

2. The formulation of the problem as a continuous optimization program is elegant. The connection established in Theorem 3, which guarantees that the solution to the practical optimization problem corresponds to the theoretically desired minimal equivalence class, is a crucial and impressive result that bridges theory and practice.


Weaknesses:

1. The paper's solution to the non-identifiability problem relies heavily on invoking the Sparsest Markov Representation (SMR) assumption. Theorem 2, which states that the sparsest graphs belong to a single MEC, is essentially a direct corollary of this assumption rather than a novel derived result. The paper expertly characterizes a problem in Theorem 1, but the principle for solving the resulting identifiability issue is assumed rather than discovered.

2. The experiments are performed only on synthetic data. While this demonstrates that DAGMA-HO works in specific cases where higher-order interactions are known to exist, the lack of any application to real-world data makes it difficult to assess the practical relevance of the problem and the utility of the proposed heuristic.


Minor: Format issues -- all appendix references are not rendered.

---

> ### Author Rebuttal · Authors · 2025-07-30
>
> We thank the reviewer for acknowledging that our paper is well‑written and that our problem formulation is elegant. We welcome the opportunity to address and clarify the reviewer’s concerns.
>
> > The paper's solution to the non-identifiability problem relies heavily on invoking the Sparsest Markov Representation (SMR) assumption. Theorem 2, which states that the sparsest graphs belong to a single MEC, is essentially a direct corollary of this assumption rather than a novel derived result. The paper expertly characterizes a problem in Theorem 1, but the principle for solving the resulting identifiability issue is assumed rather than discovered.
> >
>
> Thank you for this insightful comment. The identifiability issue arises because, with purely observational data, multiple DAGs can encode the same joint distribution—a phenomenon we formalize in Theorem 1. Without additional assumptions or interventional data, the true graph cannot be uniquely determined. While the classical faithfulness assumption addresses this, it has been shown to be overly restrictive [1]. Instead, we adopt the Sparsest Markov Representation (SMR), which is strictly weaker than faithfulness [2] and guarantees that every DAG with the minimal number of edges lies in the same Markov equivalence class. This aligns with the principle of parsimony—selecting the simplest model among those that fit the data—which is known to improve interpretability, generalization, and robustness. Notably, even without SMR, solving (13) recovers a sparsest model that exactly reproduces the observed data (Theorems 1 and 3). Furthermore, SMR is not merely theoretical: in our simulations for $p\le8$, Algorithm 3 (which exhaustively recovers all sparsest graphs, as detailed in Appendix C.4) empirically confirms that all sparsest DAGs occupy the same MEC. We will expand this discussion in the revised manuscript to clarify the role and justification of SMR.  Thanks for bring up this point to our attentions.
>
> > The experiments are performed only on synthetic data. While this demonstrates that DAGMA-HO works in specific cases where higher-order interactions are known to exist, the lack of any application to real-world data makes it difficult to assess the practical relevance of the problem and the utility of the proposed heuristic.
> >
>
> Thank you for emphasizing the need for a real‑data evaluation. We applied DAGMA‑HO to the [3] benchmark dataset, which comprises $n=7466$ continuous expression measurements of $d=11$ proteins and phospholipids in human immune cells and includes a consensus network widely accepted by the biological community. We worked with discretized dataset, and ran DAGMA‑HO. Our method recovers the consensus structure with competitive accuracy compared to existing approaches, demonstrating its potential existence of higher order interaction.
>
>
> |                | SHD | Num of Edges |
> |----------------|:---:|:------------:|
> | NOTEARS-Linear [4] |  22 |      18      |
> | NOTEARS-MLP [5]    |  16 |      13      |
> | DAGMA-HO (Ours) |  13 |      15      |
>
>
> > Minor: Format issues -- all appendix references are not rendered.
> >
>
> We apologize for these oversights and any inconvenience they may have caused. The missing references were due to compiling the paper without the appendix; a complete version without missing reference is available in the supplementary material. We will ensure that these issues do not recur in future submissions.
>
> ### Reference
>
> [1] Uhler, Caroline, et al. "Geometry of the faithfulness assumption in causal inference." *The Annals of Statistics* (2013): 436-463.
>
> [2] Raskutti, Garvesh, and Caroline Uhler. "Learning directed acyclic graph models based on sparsest permutations." *Stat* 7.1 (2018): e183.
>
> [3] Sachs, Karen, et al. ”Causal protein-signaling networks derived from multiparameter single-cell data.” Science 2005.
>
> [4] Zheng, Xun, et al. "Dags with no tears: Continuous optimization for structure learning." *Advances in neural information processing systems* 31 (2018).
>
> [5] Zheng, Xun, et al. "Learning sparse nonparametric dags." *International conference on artificial intelligence and statistics*. Pmlr, 2020.

---

> > ### Comment · Reviewer_QLcb · 2025-08-03
> >
> > I appreciate the authors' detailed response and would like to maintain my current rating.

---

> > > ### Author Response · Authors · 2025-08-03
> > >
> > > Thank you for your time and effort. If you have any further questions, we would be happy to address them.

---

### Official Review · Reviewer_coPu · 2025-07-02

**Clarity:** 3
**Significance:** 3
**Originality:** 3
**Rating:** 4
**Confidence:** 4

**Summary:**

This paper studies the problem of causal discovery in a set of discrete binary variables without any specific model. They propose a differentiable learning method that formulates the learning problem as a single differentiable optimization task. They show that general discrete models are unidentifiable from purely observational data and characterise the Markov equivalence class.

**Questions:**

Please see above.

**Ethical Concerns:**

["NO or VERY MINOR ethics concerns only"]

**Final Justification:**

I increase my score as the authors' rebuttal could address some of my concerns.

**Limitations:**

Please see above comments.

**Paper Formatting Concerns:**

References to the Appendix are missing and in some places equations are out of page border eg. (1).

**Quality:**

3

**Strengths And Weaknesses:**

The paper is well-written. The problem is a relevant and important one.
The logistic formulation of the conditional multivariate Bernoulli probabilities is quite interesting. This helps the authors to form the differentiable form and might be also useful for further researches.

There are couple of major weaknesses with this work. The contribution is limited given that the previous results on differentiable method for causal discovery such as notears.

The proposed method also suffers from the limitations of the notears like methods such as getting trapped into local minima, speed of convergence, good initialization, and the choice of parameters such as regularisation coefficient. There are no discussion about these points in the paper.

Notation B^{j_1,…,j_r} was defined for simpler representation but in some place it has not been used while it should eg. Eq. (3).

References are missing to the Appendix and in some places equations are out of page border eg. (1).

Before remark 2, “The cardinality of E(p) is at most p!”. Why not exactly p! ?
The summation in score function definition should be to n instead of p.
Function (11) is quadratic on [-delta, delta] not only on [0, delta].

---

> ### Author Rebuttal · Authors · 2025-07-30
>
> We thank the reviewer for recognizing the clarity of our presentation, for finding our work both interesting and important, and for providing many constructive comments that will help us improve the manuscript. We appreciate the opportunity to address any points of confusion and clarify our contributions.
>
> > The contribution is limited given that the previous results on differentiable method for causal discovery such as notears.
> >
>
> Thank you for raising this concern. While NOTEARS introduced an elegant differentiable acyclicity constraint that makes gradient‑based causal discovery possible, it offers **no** theoretical guarantees on its solutions and leaves unanswered what is actually learned when and how applied to discrete data. Our work fills this gap by developing a unified framework for general binary data that delivers clear theoretical properties for the estimated graph by adapting the NOTEARS formulation (see Theorem 3). Specifically, Theorem 3 shows that any optimal solution of Equation (13) recovers parameters that **exactly** reproduce the observed data distribution and yields a DAG whose conditional independencies coincide with those of the data. By modeling the **general** distribution of binary variables, we avoid the restrictive linearity assumptions of earlier methods and can capture arbitrary interactions in real‑world data. To address the exponential blow‑up in complexity inherent to general binary models, we further introduce a scalable two‑stage heuristic (NOTEARS‑MLP‑REG) that retains full modeling flexibility while remaining computationally practical on large datasets. Together, these contributions go well beyond a straightforward application of existing differentiable approaches, offering new theoretical insights, enhanced modeling power, and genuine scalability. We respectfully request the reviewer to reconsider our contributions in light of the explanations provided above.
>
> > The proposed method also suffers from the limitations of the notears like methods such as getting trapped into local minima, speed of convergence, good initialization, and the choice of parameters such as regularization coefficient. There are no discussion about these points in the paper.
> >
>
> Thank you for this helpful suggestion! Practical implementation details are indeed critical to our method. In the main text (Line 96) we briefly review related optimization‐focused studies, and in Appendix D.1 we describe our hyperparameter‐tuning procedure for the regularization coefficients. Several works have rigorously examined NOTEARS‐style methods from these implementation perspectives: [1] characterizes the KKT and local‐optimality conditions under a linear Gaussian model; [2] extends these analyses to more general settings; [3] proves that an appropriate optimization scheme converges to the global minimum in the bivariate case; and [4, 5] investigate the convergence behavior and nonconvexity of NOTEARS. Although we only mention these studies in the current submission due to space constraints, we will expand both our implementation details and our discussion of these related work in the revised manuscript.
>
> > Notation B^{j_1,…,j_r} was defined for simpler representation but in some place it has not been used while it should eg. Eq. (3).
> >
>
> Thanks! We will modify Eq. (3) in the updated version and remove the usage of $B^{j_1,…,j_r}$.
>
> > References to the Appendix are missing and in some places equations are out of page border eg. (1).
> >
>
> We apologize for these oversights and any inconvenience they may have caused. In the revised manuscript, we will split Equation (1) across multiple lines so that it fits within the page borders. The missing references were due to compiling the paper without the appendix; a complete version without missing reference is available in the supplementary material. We will ensure that these issues do not recur in future submissions.
>
> > Before remark 2, “The cardinality of E(p) is at most p!”. Why not exactly p! ?
> >
>
> We write **“at most  $p!$”** because different node orderings can induce the same DAG and parameter mapping, so not all $p!$ permutations yield distinct models. Formally, one can have $\pi_1\ne \pi_2$ with $f_{\pi_1} = f_{\pi_2}$ and $G_{\pi_1} = G_{\pi_2}$. For example, if $X = (X_1,….,X_p)$ are mutally independent any topological sort $\pi$ produces the same empty graph and identical parameters, giving $|\mathcal{E}(\mathbf{p})|=1$, rather than $p!$
>
> > The summation in score function definition should be to n instead of p.
> >
>
> We appreciate your keen attention to detail. The summation over $p$ features is indeed correct, however. In our score function $\ell(\mathbf{H};\mathbf{X}),$ the index $i$ runs over the $p$ features $X_1,\dots,X_p$, where $\mathbf{X}_i$ denotes the $i$th column of the data matrix $\mathbf X$. The sum over the $n$ observations is implicitly carried out by the inner product with the vector $\mathbf1_n^\top$ in the term that follows. We recognize that this notation can be confusing, and we will clarify it in the revised manuscript.
>
> > Function (11) is quadratic on [-delta, delta] not only on [0, delta].
> >
>
> Thank you for pointing this out. In fact, the quasi‑MCP penalty $p_{\lambda,\delta}(t)$ is an even function, symmetric about zero. On each of the subintervals $[0,\delta]$ and $[-\delta,0]$, it is given by a quadratic expression with negative curvature that attains its maximum at $|t| = \delta$. To clarify this in the revised manuscript, we will include a plot of $p_{\lambda,\delta}(t)$ and explicitly state that it is quadratic separately on $[-\delta,0]$ and $[0,\delta]$, though no single quadratic formula covers the full range $[-\delta,\delta]$.
>
>
>
> [1] Wei, et al "DAGs with No Fears: A closer look at continuous optimization for learning Bayesian networks." *Advances in Neural Information Processing Systems* 33 (2020)
>
> [2] Deng, et al. "Optimizing notears objectives via topological swaps." *International Conference on Machine Learning*. PMLR, 2023.
>
> [3] Deng, et al. "Global optimality in bivariate gradient-based DAG learning." *Advances in Neural Information Processing Systems* 36 (2023)
>
> [4] Ng, et al "Structure learning with continuous optimization: A sober look and beyond." *Causal Learning and Reasoning*. PMLR, 2024.
>
> [5] Ng, et al. "On the convergence of continuous constrained optimization for structure learning." *International Conference on Artificial Intelligence and Statistics*. Pmlr, 2022.

---

> > ### Comment · Reviewer_coPu · 2025-08-05
> >
> > I appreciate the authors' effort in addressing my comments. I will update my score accordingly.

---

> > > ### Author Response · Authors · 2025-08-05
> > > **Thank you!**
> > >
> > > Thank you for your time and effort in reviewing our paper. We truly appreciate your thoughtful feedback and constructive suggestions. We are pleased to hear that you are willing to raise your score. If there are any remaining questions or concerns that we have not fully addressed, we would be more than happy to discuss them further and provide any necessary clarifications.

---

### Official Review · Reviewer_eR7S · 2025-07-03

**Clarity:** 3
**Significance:** 3
**Originality:** 3
**Rating:** 5
**Confidence:** 5

**Summary:**

This paper presents a rigorous and well-executed study on differentiable structure learning for general binary data. Its main theoretical contribution is the proof that causal discovery is non-identifiable from purely observational data in the most general binary setting. By leveraging the multivariate Bernoulli distribution, the authors fully characterize the set of graph–parameter pairs consistent with the data, and introduce the notion of a minimal equivalence class to select among these.

The second major contribution is a novel differentiable optimization framework that recovers the causal structure up to Markov equivalence, without relying on restrictive assumptions such as linearity or additive noise. This formulation enables the recovery of the sparsest DAG consistent with the data, supported by strong theoretical guarantees.

**Questions:**

See weakness parts.

**Ethical Concerns:**

["NO or VERY MINOR ethics concerns only"]

**Final Justification:**

I believe my concern has been resolved.

**Limitations:**

My main concern is about scalability. The size of $\mathbf{H}$ matrix increase exponentially against the number of nodes in the graph (or at lease cubic if use approximation), which prevents the  algorithm to scale to even graph with near 100 nodes. Also it would be good if the authors can provide runtime comparison of different algorithms. algoriithms

**Paper Formatting Concerns:**

Minor typeset errors are noticed.

**Quality:**

3

**Strengths And Weaknesses:**

## Strengths
### Theoretical Rigor:
The paper provides a strong theoretical foundation by proving the non-identifiability of causal discovery for general binary data. It rigorously characterizes all graph–parameter pairs compatible with the data using the multivariate Bernoulli distribution.

### Novel Optimization Framework:
A differentiable formulation is introduced to recover the structure up to the Markov equivalence class, enabling the use of gradient-based methods without restrictive parametric assumptions (e.g., linear or additive noise models).

### General Applicability:
The approach handles arbitrary higher-order dependencies in binary data, overcoming limitations of existing methods that assume first-order or linear models.

### Empirical Validation:
Experiments on synthetic datasets demonstrate the method’s superiority in capturing complex causal structures, especially when higher-order interactions are present.


## Weaknesses

### Minor Writing/Typesetting Issues:
The paper appears to have been written in a rush. There are several typesetting and referencing errors—for example, broken or missing references to appendices (e.g., “Appendix ??”), which slightly detract from readability and polish.


### Limited Empirical Analysis of Optimization Aspects (Minor):
While the authors adopt the DAGMA framework for optimization, the analysis of the DAG constraints themselves is underdeveloped. The paper could be strengthened by a more detailed empirical or theoretical investigation of how the chosen DAG constraint affects optimization performance and recovery quality.

In particular, recent work suggests that the acyclicity constraint in DAGMA could be replaced with:

- Matrix inversion-based constraints [1,2], if vanishing gradients are a concern.

- NOTEARS-style constraints, if non-convexity of the objective is the dominant challenge.

Incorporating or comparing such alternatives could improve the overall robustness and insight of the paper’s optimization component.

[1] Zhang, Zhen, et al. "Truncated matrix power iteration for differentiable DAG learning." Advances in Neural Information Processing Systems 35 (2022): 18390-18402.
[2] Zhang, Zhen, et al. "Analytic DAG Constraints for Differentiable DAG Learning." arXiv preprint arXiv:2503.19218 (2025).

---

> ### Author Rebuttal · Authors · 2025-07-30
>
> We thank the reviewer for acknowledging that our paper offers a rigorous theoretical framework for general binary data—capable of modeling arbitrary higher‑order interactions—as well as a novel optimization framework. We appreciate this recognition and welcome the opportunity to address any remaining concerns.
>
> > Minor Writing/Typesetting Issues: .... readability and polish.
> >
>
> We apologize for this oversight and any inconvenience it may have caused during the review. The issue arose because the manuscript was compiled without the appendix content. A clean version—including all appendices—is available in the supplementary material, and we will ensure this does not happen again. Thank you for bringing it to our attention.
>
> > Limited Empirical Analysis of Optimization Aspects
> >
> Thank you for highlighting these foundational works. Our method’s success hinges on how we enforce the acyclicity constraint in optimization (13), both theoretically and numerically. In the revised manuscript, we will add a dedicated discussion of prior approaches to DAG constraints—particularly the theoretical contributions of [1,2]—and report new experiments in which we replace our DAGMA constraint with the acyclicity formulation from [1,2], presenting those comparative results.
>
> Below, we report the runtime and SHD when the exponential acyclicity constraint is replaced by the method of [1], yielding our variant NOTEARS‑MLP‑REG‑TMPI. The source code for [2] is not yet publicly available; we will perform a similar comparison once it is released on GitHub. These results demonstrate that incorporating the enhanced acyclicity constraints from [1,2] can further improve our method’s performance. Thanks for pointing out this.
>
>
> | Times(second) | NOTEARS-MLP-REG(Ours)| NOTEARS-MLP-REG-TMPI |
> |---------------|:---------------:|:--------------------:|
> | ER2 (d=40)    |     295.333     |        272.389       |
> | ER4 (d=40)    |     337.205     |        301.232       |
> | SF2 (d=40)    |     293.903     |        273.130       |
> | SF4 (d=40)    |     313.349     |        292.179       |
>
> | SHD        | NOTEARS-MLP-REG(Ours)| NOTEARS-MLP-REG-TMPI |
> |------------|:---------------:|:--------------------:|
> | ER2 (d=40) |      42.323     |        43.172        |
> | ER4 (d=40) |      73.323     |        67.293        |
> | SF2 (d=40) |       48.2      |        43.890        |
> | SF4 (d=40) |      103.2      |        92.333        |
>
> > My main concern is about scalability..... Also it would be good if the authors can provide runtime comparison of different algorithms.
> >
>
> We appreciate the reviewer’s concern about scalability. As noted in Lines 314–322, the size of our feature map $\Phi(\mathbf X)$ (and thus the cost of solving (13)) grows exponentially with the number of variables $p$, so a direct solution quickly becomes infeasible for even moderately large graphs. To address this, we introduce a two‑stage heuristic called NOTEARS‑MLP‑REG. In the first stage, we train an MLP (NOTEARS‑MLP[3]) to capture higher‑order interactions in discrete data and recover an initial graph structure. From this preliminary graph we extract a node ordering $\hat{\pi}$, which we then use in the second stage to fit a logistic‑regression model—including both first‑ and second‑order terms—to obtain the final graph. Because logistic regression under a fixed ordering is very efficient, the MLP‑based graph recovery remains the dominant computational cost. The runtime of our method (NOTEARS‑MLP-REG) is quite similar to NOTEARS‑MLP[3].
>
> The following table provides the runtime for different methods and will be included in the final version. Here,  DAGMA is renamed as “DAGMA-1st”, to emphasize only linear term is used.
>
>
> |Times(second)| NOTEARS-MLP-REG(Ours) | DAGMA-1st | FGES   | PC     |
> |------------ |:---------------:|:---------:|:------:|:------:|
> | ER2 (d=10)  | 43.223          | 11.291    | 2.2565 | 3.3855 |
> | ER4 (d=10)  | 38.972          | 12.887    | 2.3755 | 5.522  |
> | SF2 (d=10)  | 37.925          | 12.789    | 2.192  | 3.3545 |
> | SF4 (d=10)  | 41.259          | 11.567    | 2.658  | 4.4015 |
>
> |Times(second)| NOTEARS-MLP-REG(Ours) | DAGMA-1st | FGES   | PC      |
> |-------------|:---------------:|:---------:|:------:|:-------:|
> | ER2 (d=20)  | 84.392          | 26.291    | 3.7165 | 6.6575  |
> | ER4 (d=20)  | 87.267          | 28.987    | 6.2005 | 14.349  |
> | SF2 (d=20)  | 80.952          | 25.781    | 3.7605 | 6.5365  |
> | SF4 (d=20)  | 93.136          | 28.659    | 5.6295 | 11.7385 |
>
> |Times(second)| NOTEARS-MLP-REG(Ours) | DAGMA-1st |  FGES  |  PC    |
> |-------------|:---------------:|:---------:|:------:|:------:|
> | ER2 (d=40)  |     295.333     |   43.232  | 4.2745 |  14.49 |
> | ER4 (d=40)  |     337.205     |  58.2323  | 7.6775 | 37.806 |
> | SF2 (d=40)  |     293.903     |   42.232  |  4.215 |  23.11 |
> | SF4 (d=40)  |     313.349     |  48.6789  | 11.616 |  21.21 |
>
> From the reported runtimes, it is clear that our method does not exhibit exponential time complexity and can deliver meaningful discoveries within a reasonable timeframe.
>
> __Reference__
>
> [1] Zhang, Zhen, et al. "Truncated matrix power iteration for differentiable DAG learning." Advances in Neural Information Processing Systems 35 (2022)
>
> [2] Zhang, Zhen, et al. "Analytic DAG Constraints for Differentiable DAG Learning." arXiv preprint arXiv:2503.19218 (2025).
>
> [3] Zheng, Xun, et al. "Learning sparse nonparametric dags." *International conference on artificial intelligence and statistics*. Pmlr, 2020.

---

### Comment · Area_Chair_iR13 · 2025-08-01
**The time to start author-reviewer discussions**

Dear all reviewers,

The author rebuttal period has now concluded, and authors' responses are
available for the papers you are reviewing. The Author-Reviewer Discussion
Period has started, and runs until August 6th AoE.

Your active participation during this phase is crucial for a fair and
comprehensive evaluation. Please take the time to:

- Carefully read the author responses and all other reviews.
- Engage in a constructive dialogue with the authors, clarifying points,
  addressing misunderstandings, and discussing any points of disagreement.
- Prioritize responses to questions specifically addressed to you by the authors.
- Post your initial responses as early as possible within this window to
  allow for meaningful back-and-forth discussion.

Your insights during this discussion phase are invaluable.
Thank you for your continued commitment to the NeurIPS review process.

Bests,
Your AC

---

### Decision · Program_Chairs · 2025-09-17

**Decision:**

Accept (poster)

**Comment:**

This paper introduces a differentiable structure learning framework for
general binary data, moving beyond the restrictive parametric assumptions
of prior work to capture arbitrary higher-order interactions. The work is
grounded in a rigorous theoretical analysis, starting from the general
multivariate Bernoulli distribution to first prove non-identifiability and
then characterise the complete set of compatible causal structures.

The reviewers initially identified this as a borderline case, raising
several valid concerns. The primary issues included the method's
exponential complexity and questionable scalability, an evaluation limited
to synthetic data, and the reliance on the Sparsest Markov Representation
(SMR) assumption for identifiability.

However, the authors provided a comprehensive and convincing rebuttal that
successfully addressed these key points. They clarified the practical
performance of their proposed heuristic with detailed runtime comparisons,
demonstrating its feasibility. Crucially, they added a new experiment on a
well-known real-world biological dataset, which provided strong evidence of
the method's practical utility and directly addressed a major limitation.
The authors also effectively justified their use of the SMR assumption and
committed to improving the paper's clarity. The productive discussion and
the authors' substantive responses led multiple reviewers to increase their
scores or strengthen their support for the paper.

Given the strong theoretical foundation, the novelty of addressing
structure learning in a general binary setting, and the new empirical
evidence demonstrating its real-world applicability, the paper now stands
as a solid contribution. Hence the AC recommend acceptance.